# Statistical Analysis of Common Respiratory Viruses Reveals the Binary of Virus-Virus Interaction

Lulu Zhang,[a] Yan Xiao,[a,b] Zichun Xiang,[a] Lan Chen,[a] Ying Wang,[a] Xinming Wang,[a] Xiaojing Dong,[c] Lili Ren,[a,b] Jianwei Wang[a,b]

[a]Institute of Pathogen Biology, Chinese Academy of Medical Sciences & Peking Union Medical College, Beijing, People's Republic of China
[b]Key Laboratory of Respiratory Disease Pathogenomics, Chinese Academy of Medical Sciences and Peking Union Medical College, Beijing, People's Republic of China
[c]Santa Clara University, Santa Clara, California, USA

Lulu Zhang and Yan Xiao contributed equally to this work as first authors. The order of co-author was determined in order of increasing effort on manuscript preparation.
Xiaojing Dong, Lili Ren, and Jianwei Wang contributed equally to this work as senior authors.

**ABSTRACT** Respiratory viruses may interfere with each other and affect the epidemic trend of the virus. However, the understanding of the interactions between respiratory viruses at the population level is still very limited. We here conducted a prospective laboratory-based etiological study by enrolling 14,426 patients suffered from acute respiratory infection (ARI) in Beijing, China during 2005 to 2015. All 18 respiratory viruses were simultaneously tested for each nasal and throat swabs collected from enrolled patients using molecular tests. The virus correlations were quantitatively evaluated, and the respiratory viruses could be divided into two panels according to the positive and negative correlations. One included influenza viruses (IFVs) A, B, and respiratory syncytial virus (RSV), while the other included human parainfluenza viruses (HPIVs) 1/3, 2/4, adenovirus (Adv), human metapneumovirus (hMPV), and enterovirus (including rhinovirus, named picoRNA), $\alpha$ and $\beta$ human coronaviruses (HCoVs). The viruses were positive-correlated in each panel, while negative-correlated between panels. After adjusting the confounding factors by vector autoregressive model, positive interaction between IFV-A and RSV and negative interaction between IFV-A and picoRNA are still be observed. The asynchronous interference of IFV-A significantly delayed the peak of $\beta$ human coronaviruses epidemic. The binary property of the respiratory virus interactions provides new insights into the viral epidemic dynamics in human population, facilitating the development of infectious disease control and prevention strategies.

**IMPORTANCE** Systematic quantitative assessment of the interactions between different respiratory viruses is pivotal for the prevention of infectious diseases and the development of vaccine strategies. Our data showed stable interactions among respiratory viruses at human population level, which are season irrelevant. Respiratory viruses could be divided into two panels according to their positive and negative correlations. One included influenza virus and respiratory syncytial virus, while the other included other common respiratory viruses. It showed negative correlations between the two panels. The asynchronous interference between influenza virus and $\beta$ human coronaviruses significantly delayed the peak of $\beta$ human coronaviruses epidemic. The binary property of the viruses indicated transient immunity induced by one kind of virus would play role on subsequent infection, which provides important data for the development of epidemic surveillance strategies.

**KEYWORDS** acute respiratory infection, etiology, epidemiology, surveillance, statistic model, virus-virus interaction

Address correspondence to Jianwei Wang, wangjw28@163.com, Lili Ren, renliliipb@163.com, or Xiaojing Dong, xdong1@scu.edu.

The authors declare no conflict of interest.

Acute respiratory infection (ARI) is the major cause of morbidity and mortality worldwide. Each person would be attacked 2 to 3 episodes of ARI each year (1–3) and a quarter of the population would need primary care (4). Respiratory viruses are the most common agents

of ARI, including influenza viruses (IFVs), respiratory syncytial virus (RSV), human coronaviruses (HCoVs, including 229E, OC43, NL63 and HKU1 before COVID-9 pandemic), parainfluenza virus (HPIV)1 to 4, enterovirus (EVs)/human rhinovirus (HRVs), adenovirus (Adv), human metapneumovirus (hMPV), human bocavirus (HBoV) (5, 6). These viruses have different epidemic patterns, population susceptibility, and mechanisms of infection (6–9). The ecology of respiratory tract depends on the dynamic regulation among microbes and host. At present, the viruses-host, bacteria-host, and viruses-bacteria interactions have been widely studied (10–13), but the research on virus-virus interactions is relatively little. Elucidating this gap in knowledge will help prevent and treat respiratory infectious diseases.

Viral interference has been recognized and reported previously. During the 2009 pandemic of an emerging IFV-A, when data from several European countries and our group indicated that the annual autumn HRVs epidemic interrupted and delayed transmission of the emerging IFVs (13–15). Studies from the United Kingdom provided the first systematic large-scale presentation of respiratory virus interactions (16). Clinical studies and experimental animal models also provided initial evidence of viral interactions at the host level (17). However, the mechanism behind virus-virus interactions is elusive. Adaptive immunity is one of the possible explanations (17, 18). Cross-immunity caused by infected with one virus can alter the epidemic dynamics of another virus (19). Other possible explanations include competition for resources and other biological processes (20).

The profile of virus-virus interaction is still rarely reported. The reason is that the limited coepidemic data involving all the common respiratory viruses restrict the understanding on interactions between the respiratory viruses over a long period. And it is difficult to explain whether the phenomenon is etiology driven or host-variable driven even data are available.

In this study, we conducted a prospective laboratory-based respiratory virus etiological study by enrolling patients suffered from ARI in Beijing, China during 2005 to 2015. Each enrolled patient was simultaneously tested for 18 common respiratory viruses. We quantitatively assessed viral correlations based on time series data. Viral interactions were characterized after excluding confounding factors. In addition, taking HCoVs as an example, we identified the effect of viral interactions on virus detections, which provides ecological insights into the viral transmission dynamics in human population and informs the development of predictive model of infectious diseases control as well as intervention strategies.

## RESULTS

**Detection of respiratory viruses.** A total of 14,426 adult outpatients with ARI were enrolled in our study, in which 963 (6.68%) aged ≥65 years, and 954 (6.61%) had an underlying disease during visiting. The most common underlying disease was hypertension ($n = 306$, 2.12%), followed by chronic liver, heart, or renal diseases ($n = 196$, 1.36%), diabetes ($n = 144$, 1.00%), cancer ($n = 133$, 0.92%), and chronic lung diseases ($n = 83$, 0.58%). The elderly patients had the most underlying diseases ($n = 267/963$, 27.73%, $P < 0.001$, Chi-square test) among all age groups. Self-administered antibiotics usage before visiting was reported in 2,082 (14.43%) patients. Antivirals usage before visiting was reported in 26 (<1%) (Table S1).

At least one virus was detected in 5,585 patients (38.71%), in which 5,243 (36.34%) were single-detected and 342 (2.37%) were multiple detected with two or more viruses. The most frequent single-detected virus was IFV-A ($n = 2,399$, 16.63%), followed by IFV-B ($n = 814$, 5.64%), HRVs ($n = 798$, 5.53%), EVs ($n = 355$, 2.46%), and Adv ($n = 151$, 1.05%). The detection rates of other 13 respiratory viruses were less than 1% (Table S2). HRVs ($n = 173$, 50.58%), IFV-A ($n = 165$, 48.25%), and IFV-B ($n = 73$, 21.35%) were the most frequently detected viruses in the cases with multiple pathogens codetected. IFV-A with HRVs ($n = 83$, 24.27%) were the most common dual-detected viruses (Table S3).

The overall detection rates of respiratory viruses varied monthly with a range of 0% to 86.67% (Fig. 1A). According to the mainly peaked season, the 18 respiratory viruses were subjectively grouped into four groups (Fig. 1B). Group I included hMPV, Adv and HPIV-4, mainly detected in spring. Group II included HPIVs 1 to 3, EVs, HCoV-OC43, and HCoV-HKU1, mainly detected in summer. Group III included HRVs, HBoV, HCoV-229E, and HCoV-NL63, which were in slightly higher detection rates in autumn. Group IV included IFVs and RSV,

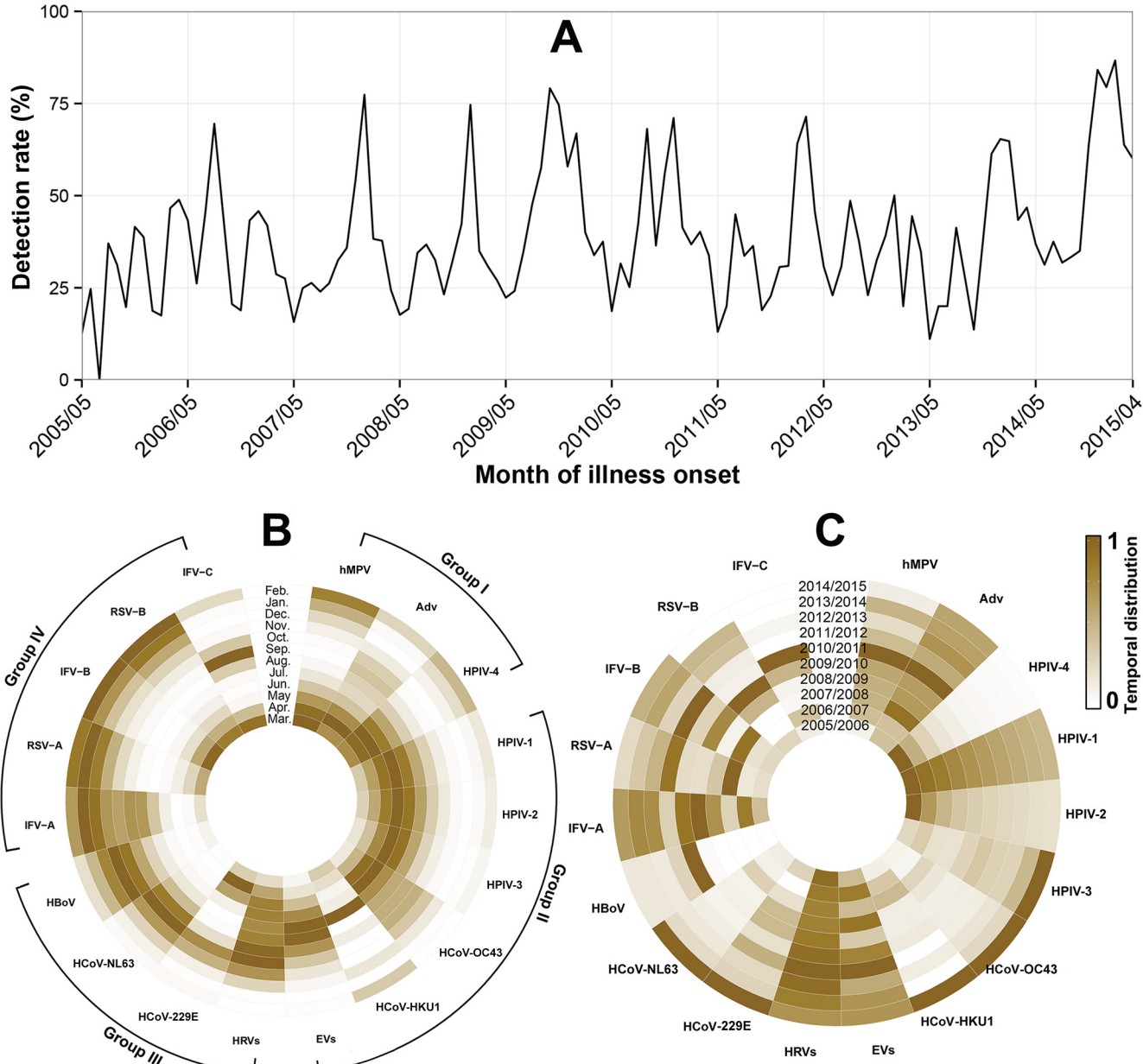

**FIG 1** Temporal distribution of viral pathogens in adults with acute respiratory infections in Beijing, China, 2005 to 2015. (A) Overall monthly detection rate of respiratory viruses. (B) Thermodynamic diagram of average monthly detection rate, scaled from 0 to 1 according to percentile rank, by viruses. (C) Thermodynamic diagram of yearly detection rate, scaled from 0 to 1 according to percentile rank, by viruses. IFVs (A, B, C), influenza virus (type A, B, C); HRVs, human rhinoviruses; HPIVs (1–4), human parainfluenza viruses (type 1, 2, 3, 4); EVs, enteroviruses; Adv, adenoviruses; HCoVs (NL63, HKU1, OC43, 229E), human coronaviruses (type NL63, HKU1, OC43, 229E); RSV (A, B), respiratory syncytial virus (subgroup A, B); hMPV, human metapneumovirus; HBoV, human bocaviruses.

mainly detected in winter. The detection of IFV-A, Adv, EVs, and hMPV showed significant periodicity ($P < 0.05$) when analyzed by using wavelet method, while IFV-B, RSV, HPIVs, HRVs, and HCoVs showed no significant epidemic periodicity ($P > 0.05$) (Fig. 1C, Fig. S1).

The detection rate of respiratory virus showed different among age groups ($P = 0.023$, Chi-square test), with the highest rate in those 14 to 24 years old (40.69%) and lowest in those ≥65 years old (37.69%) (Table S2). HRVs, EVs and Adv were more frequently detected in those aged 14 to 24 years, IFVs in those aged 45 to 64 years, with RSV, HPIVs, and HCoVs more frequent in elderly adults ($P < 0.05$, Chi-square test) (Fig. 2, Table S2).

**Correlations between respiratory viruses.** To test whether there exist interactions between respiratory viruses, we analyzed the correlations between respiratory viruses by using Spearman's rank correlation test. The data showed seven negative correlations and

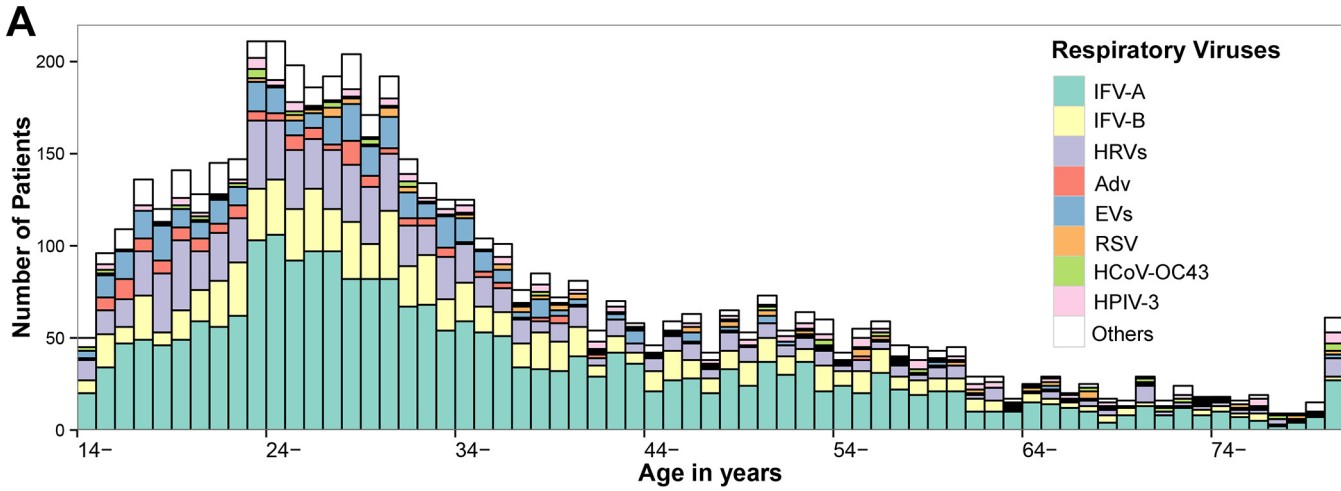

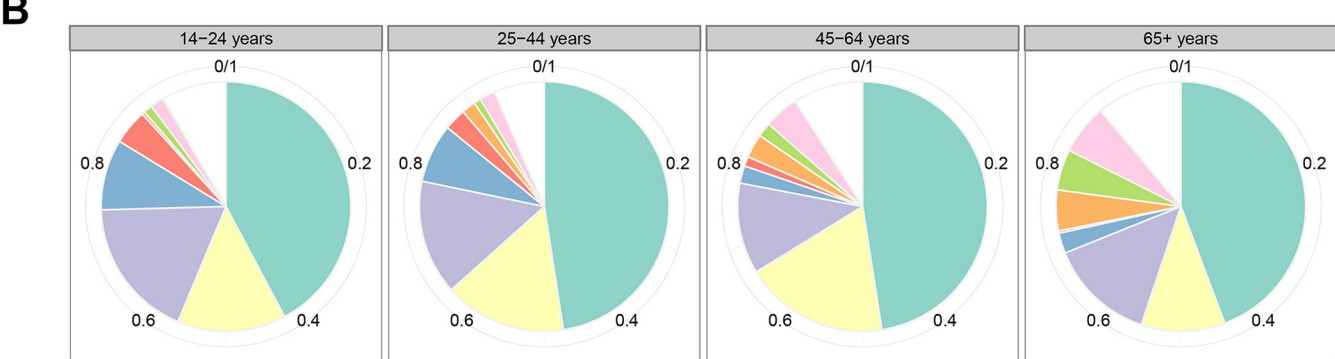

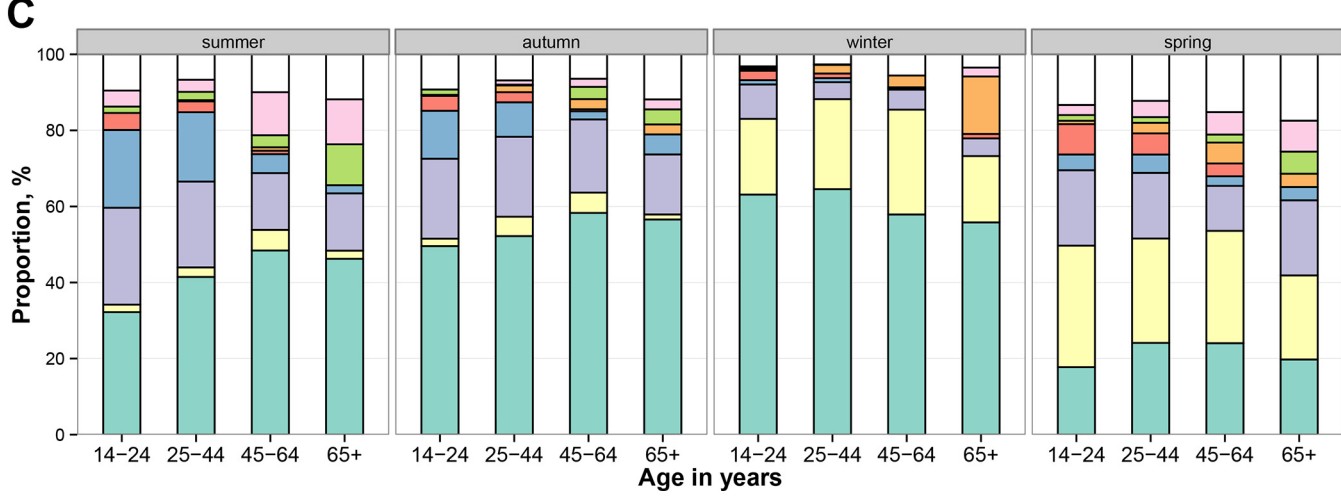

**FIG 2** Age distribution of viral pathogens in adults with acute respiratory infections in Beijing, China, 2005 to 2015. (A) No. of detections, by viruses. (B) Proportions of detections, by viruses. (C) Proportions of detections, by viruses and illness onset season.

seven positive correlations among the viruses detected (Fig. 3A). The positive correlations were found between RSV with IFV-B ($r = 0.56$, $P < 0.001$), HPIV 1/3 with HPIV 2/4 ($r = 0.35$, $P < 0.001$), HCoVs-$\beta$ with HPIV 2/4 ($r = 0.31$, $P = 0.001$), Adv with HCoVs-$\alpha$ ($r = 0.21$ $P = 0.021$), Adv with hMPV ($r = 0.30$, $P = 0.001$), HPIV 2/4 with picoRNA ($r = 0.22$, $P = 0.015$) and HPIV 1/3 with picoRNA ($r = 0.24$, $P = 0.009$). Negative correlations were observed between IFV-A with picoRNA ($r=-0.23$, $P = 0.012$), IFV-B with picoRNA ($r=-0.33$, $P < 0.001$), RSV with picoRNA ($r=-0.29$, $P = 0.001$), HPIV 2/4 with IFV-A ($r=-0.23$, $P = 0.013$), HPIV 2/4 with RSV ($r=-0.25$, $P = 0.006$), HPIV 1/3 with IFV-A ($r=-0.22$, $P = 0.016$) and HPIV 1/3 with RSV ($r=-0.20$, $P = 0.031$) (Fig. 3A, Fig. S2).

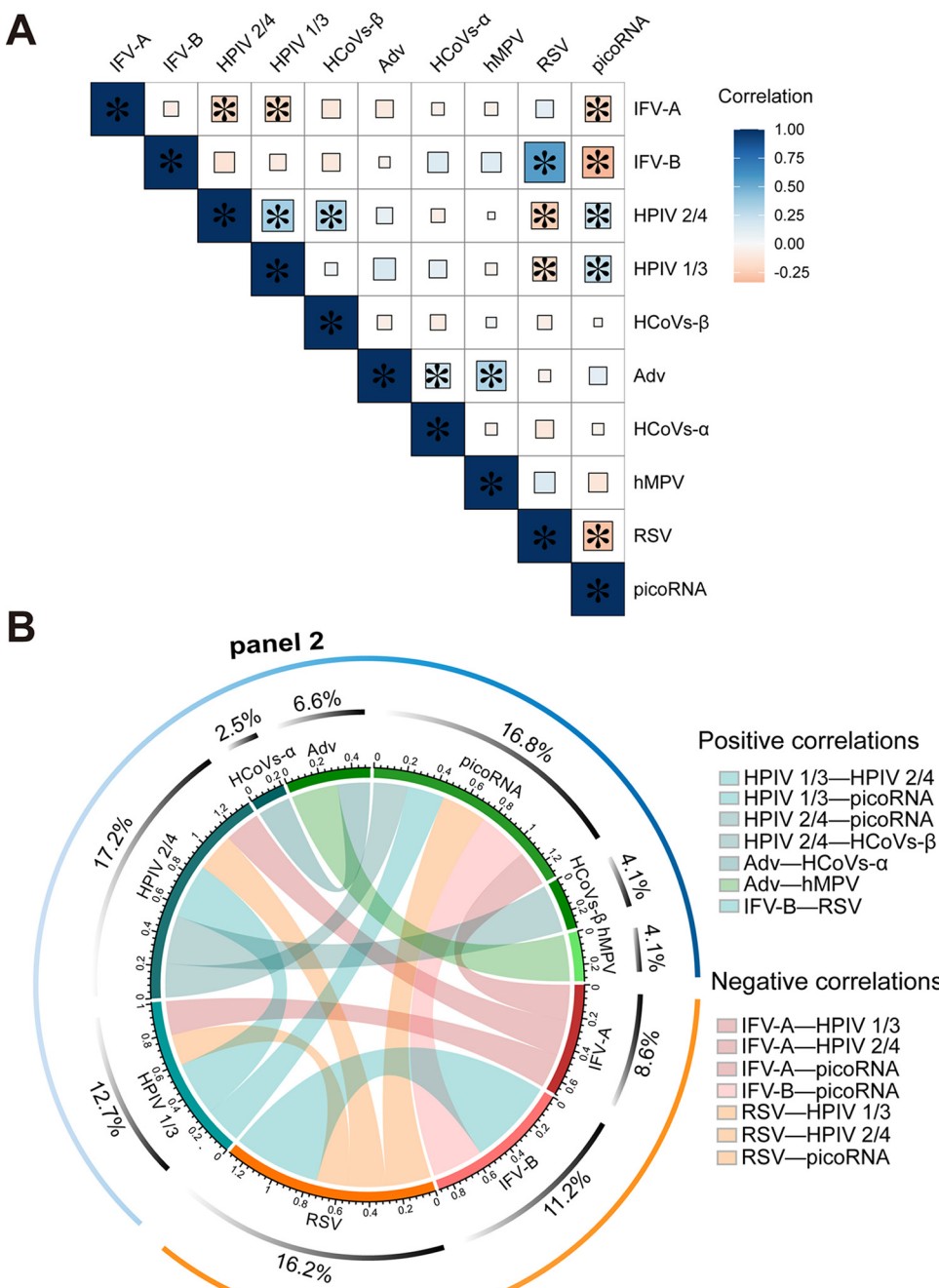

**FIG 3** Correlations of detection rate between respiratory viruses. (A) Correlations among respiratory viruses at the human population level. The heatmap represents correlations between common respiratory viruses, in which blue squares represent positive correlations and red squares represent negative correlations. The larger the square, the darker the color and the higher the correlation. Correlations with *P* values of statistical tests less than 0.05 were marked with an asterisk. (B) Virus correlations network of common respiratory viruses. The blue and green strips represent positive correlations, and the orange and rose strips represent negative correlations. The thickness of the strip represents the size of the correlations. The ability of each virus to make connections with other viruses is marked in the middle circle. All these correlations were statistically significant (Spearman's rank correlation test, *P* < 0.05). IFV-A, influenza A virus; IFV-B, influenza B virus; HPIV 1/3, human parainfluenza viruses (type 1, 3); HPIV 2/4, human parainfluenza viruses (type 2, 4); HCoVs-α, human coronaviruses (type NL63, 229E); HCoVs-β, human coronaviruses (type OC43, HKU1); hMPV, human metapneumovirus; Adv, adenoviruses; RSV (A, B), respiratory syncytial virus (subgroup A, B); picoRNA, picornaviridae (including human rhinoviruses and enteroviruses).

Based on the combinations of positive and negative correlated viruses, the respiratory viruses could be further grouped into two panels. One panel included IFV-A, IFV-B, and RSV, while the other included HPIV 1/3, HPIV 2/4, HCoVs-$\alpha$, HCoVs-$\beta$, Adv, hMPV, and picoRNA. Within each panel, all the viruses showed positive correlations, while negative correlations were found between the two panels (Fig. 3B). As exemplified, IFV-A, IFV-B, and RSV, the main pathogens of respiratory infections in winter, showed negative correlations with all the other tested viruses in our study.

**Viral interactions estimated by vector autoregressive model.** We further evaluated the stability of the observed correlations between viruses across 10 years. Although the total number of samples varies from year to year, the distribution of age group is similar every year (Fig. S3). It showed that the virus correlations fluctuated between $-0.50$ and $0.50$ without zero during 2005 to 2015, indicating the virus correlations were stable (Fig. 4A).

To distinguish genuine virus interactions from simple correlations, we performed vector autoregressive model analysis controlling for confounding factors, including age, gender, and season. The model focuses on effects from the lag of the pathogen itself and the impulse response and lag of other viral variables. The unit root test showed that all 10 of the virus sequences were stationary, and when the lag phase was equal to 2, the Akaike information criterion (AIC) was the minimum ($-69.978$), and the model fit best. According to the correlation matrix of residuals, there are positive interaction between IFV-A and RSV (correlation 0.047), and negative interaction between IFV-A and picoRNA (correlation $-0.055$) (Fig. 4B, Table S4 and 5). Orthogonal impulse response analysis showed that the pulse from IFV-A had a positive effect on RSV and a negative effect on picoRNA. The effect was maintained to lag phase 10, indicating the far-reaching effect of viral interactions (Fig. 4C).

**Delaying effect of viral interactions on the epidemic of $\beta$ human coronaviruses.** During the study period, the epidemic of HCoVs-$\beta$ (HCoV-OC43 and HCoV-HKU1) was at a relative low level and mainly peaking in summer. A significant asynchronous interference between IFVs and HCoVs-$\beta$ was identified in spring and summer (spring: r=$-0.04$, $P = 0.029$; summer: r=$-0.04$, $P = 0.009$) (Fig. 5A and B). We used Susceptible-Infectious-Recovered (SIR) model to simulate the influence of asynchronous interference on the epidemic of HCoVs-$\beta$. It showed that the peaking time of HCoVs-$\beta$ would be delayed from original 120 days to 130 days in spring and 135 days in summer if asynchronous interference was involved (Fig. 5C and D). This model was then used to simulate the effect of viral interference on the epidemic of $\beta$ human coronaviruses with different basic transmission rates (R0 = 2.70, 5.50, and 8.00). It showed that the impact of asynchronous interference of IFVs on the epidemic activity of $\beta$ human coronaviruses decreased with the increasing of R0 (Fig. 5E and F). When R0 reaching to 8.00, the asynchronous interference of IFVs had no significant effect on the peaking time of $\beta$ human coronaviruses.

## DISCUSSION

In this study, we characterized the distribution and interactions of common respiratory viruses from 2005 to 2015, based on a laboratory-based pathogen screening on patients suffered from ARI. We found there exists stable correlations between respiratory viruses across 10 year period. IFVs combined with RSV, showed negative correlations with other common respiratory viruses. IFVs showed asynchronous interference with $\beta$ human coronaviruses by delaying the duration of the epidemic peak.

Virus interaction is an important interfering factor affecting the epidemic magnitude, incidence and peak of respiratory pathogens (13, 21, 22). Viral transmissibility, human exposure history, and the cross immunity induced by the infected viruses were the potential mechanisms on virus interactions (21, 22). Jenner first reported in 1804 that herpes infections may prevent the development of vaccinia lesions, which was named viral interference. Till now, the interactions among viruses and the effects on viral epidemic were still not well defined. Among viruses with high epidemic activity, such as IFVs, RSV and HPIVs, neutral or competitive effects among the viruses were still in debated considering the population distribution, individual morbidity, and laboratory investigations (13, 22). One of the major reasons is the lack of epidemiological data, the absent of mathematical models and integrated analysis.

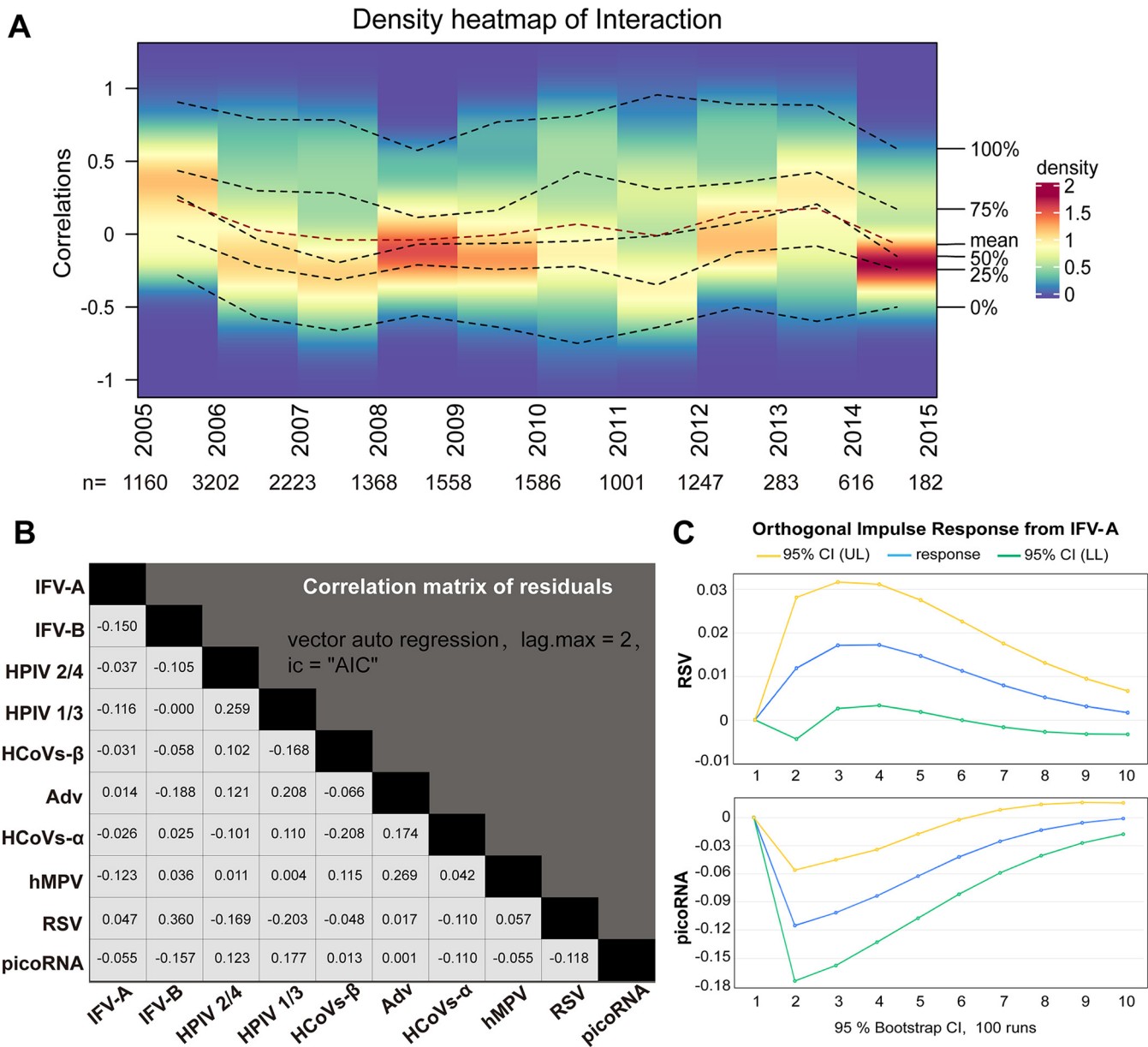

**FIG 4** Viral interactions estimated by vector autoregressive model. (A) The temporal density map of correlations of virus detections from 2005 to 2015. The *y* axis on the left representing the Spearman correlation coefficients by year. The percentages shown on the right side of the heat map represent the mean, median, quartile and range of the correlation coefficients, and the change in these values from year to year was represented by a six dotted lines. The density bar on the right used color to map the density value of the distribution. The darker the color, the greater the density. (B) Correlation matrix of residuals for viruses based on vector autoregressive model. Lag = 2, endogenous variables are 10 viruses and exogenous variables included age, gender and season. (C) Orthogonal impulse response from IFV-A to RSV and picoRNA. The horizontal axis represents the number of lag periods, the vertical axis represents the size of the response, the blue lines represent the impulse response function, and the yellow and green lines represent the 95% CI error deviation band.

Our study integrated laboratory-based viral screening data across 10 years, demographic factors, and statistics models. We found all the tested viruses showed long-term and stable interactions. The asynchronous interference between IFV-A and HRVs found in this study has also been reported in other studies (16, 17), although studies were conducted in different populations, suggesting that transient immunity from IFV-A infection is effective against HRVs. In addition, we found significant negative interference between IFVs and other viruses, which indicated that the immune response against IFVs infection may also induced cross-immune protection against other viruses. Such findings will be important for the development of vaccine strategies.

In our study, the peaking time of HCoVs-β was mainly in summer, with 7-fold frequencies compared with that of winter. Significant negative correlations between IFV-A and HCoVs-β

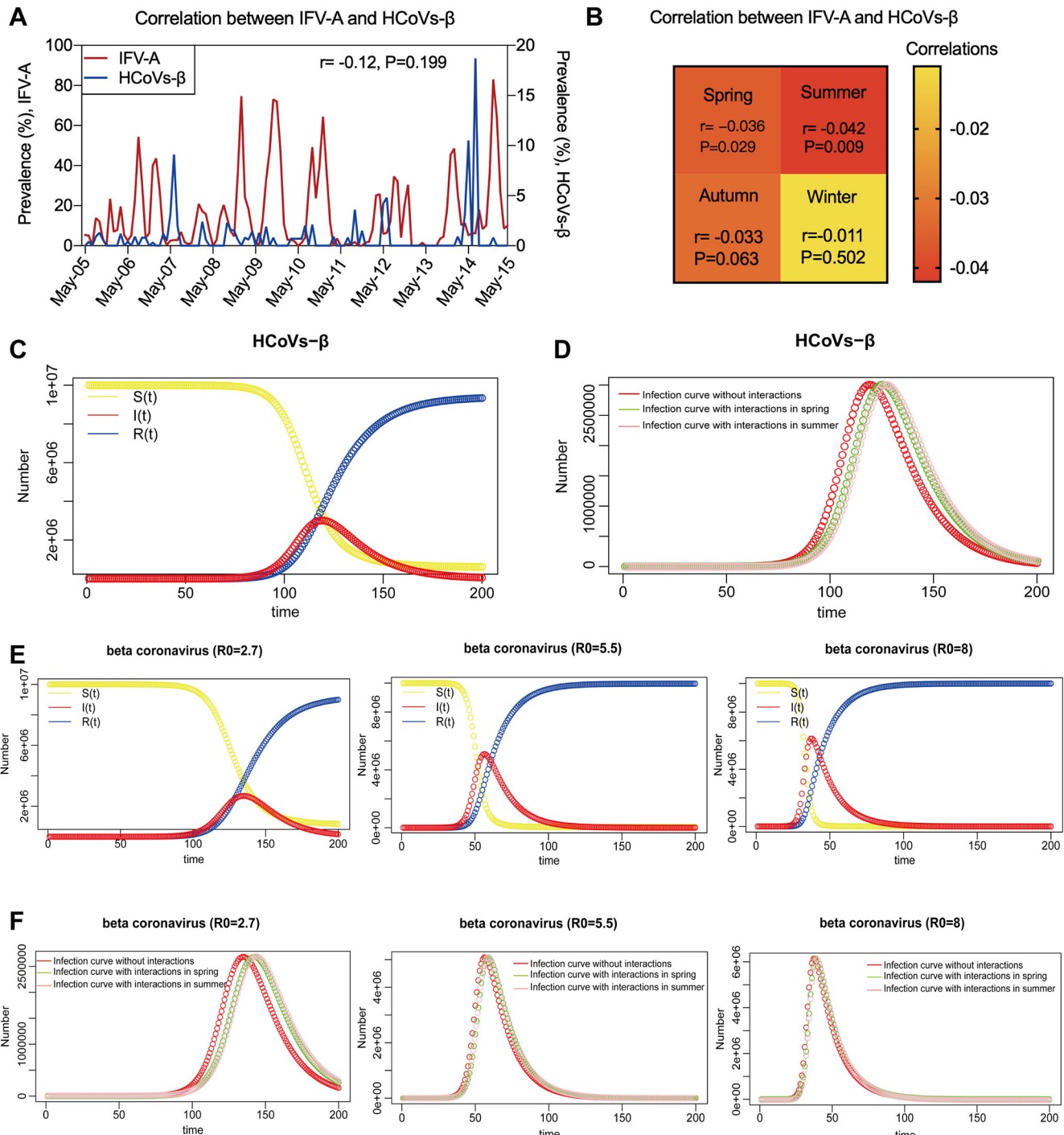

**FIG 5** The Dynamic model simulates the effect of viral interactions on $\beta$ human coronaviruses detection. (A) The detection of IFV-A showed no correlations with HCoVs-$\beta$ (HCoV-OC43 and HCoV-HKU1) by using Spearman's rank correlation test. (B) The interaction between IFV-A and HCoVs-$\beta$ was significant in spring and summer, but not in autumn and winter. (C) SIR (Susceptible-Infectious-Recovered) models simulate transmission dynamic of HCoVs-$\beta$. (D) The model predicts the impact of IFV-A asynchronous interference on HCoVs-$\beta$ epidemic. (E) SIR models simulate the transmission dynamic of $\beta$ human coronaviruses with different transmission rates (R0 = 2.7, 5.5, and 8.0). (F) The model predicts the impact of IFV-A asynchronous interference on $\beta$ human coronaviruses (R0 = 2.7, 5.5, and 8.0) epidemic.

were observed in spring and summer, but not in autumn and winter. The low detection rate may have influenced the interaction analysis. In addition, it has limited effect on viruses with high transmission rate, which suggested that we should be more vigilant against coinfection of high transmissible strains with IFVs. Although our study used the simplest SIR model, these results provide important reference data for epidemic prevention and control in the post-COVID-19 era, which needs to pay attention to coinfection of SARS-CoV-2 and IFVs (23, 24).

In addition, we combined virus as HPIV 1/3, HPIV 2/4, picoRNA, HCoVs-$\alpha$, and HCoVs-$\beta$ according to the taxonomy characteristics as some virus subtypes have detection rate lower than 1%. The combining analysis enabled us to investigate the interactions between the species of viruses; however, the interactions of viral subtypes might be ignored.

There are some limitations in our study. First, the study is a hospital-based investigation, while our findings might be further confirmed by involved much more data from the community. Second, children's cases are not included, and our findings should be further validated in a larger population. Finally, our study was conducted in the north of China and the characteristics of virus interactions need to be evaluated intensively by involving more data from other geographical regions.

In summary, our study systematically characterized the interactions between respiratory viruses through statistical model based on the data of multipathogen detections in large scale samples and revealed the binary of viral interactions. These results will improve our understanding on the epidemic of respiratory viruses. Further understanding of the biological mechanisms underlying viral interactions could help improve disease prediction and evaluation policies in the future.

## MATERIALS AND METHODS

**Study design.** From May 1, 2005, through April 30, 2015, patients suffering from ARI aged no less than 14 years were enrolled at the Fever Outpatient Clinic Department (FOCD) in Peking Union Medical College Hospital, Beijing, China. The first 5 to 10 patients who met the criteria each day were enrolled during the study period and the remaining patients were not included. The recruitment criteria included (1) acute onset; (2) fever (body temperature ≥38℃); (3) normal or low leukocyte count (a white blood cell count ≤ 10,000/mL); (4) symptoms or signs of respiratory tract infections. Patients were excluded if they had been hospitalized 1 week previously, where aged less than 14 years, or had been enrolled in the study within the previous 28 days. Demographic and clinical data were collected from each patient by using a standardized case reporting form. For each enrolled patient, nose and throat swabs were collected and the two swabs were pooled in one tube containing virus transport medium (VTM; Copan, Brescia, Italy).

**Respiratory virus screening.** Total nucleic acid (DNA and RNA) was directly extracted from each specimen by using the NucliSens easyMAG apparatus (bioMérieux, Marcy l'Etoile, France), according to the manufacturer's instructions. The presence of IFVs A, B and C, HPIVs 1 to 4, RSV A and B, Adv, HBoV, hMPV, EVs (including HRVs), and HCoV-229E, HCoV-OC43, HCoV-NL63, and HCoV-HKU1 were screened by using reverse transcription-PCR (RT-PCR) and PCR method. Detailed information on primers and laboratory procedures has been summarized previously (6).

**Statistical analysis.** The detection rate of virus was calculated by dividing the number of positive cases by the total tested number. The proportions of positive detections were compared by using Chi-square or Fisher's exact tests. A two-sided $P$ value of <0.05 was considered statistically significant. The monthly distribution was scaled from 0 to 1 of each virus, according to the percentile rank. Four seasons were determined as spring (March-May), summer (June-August), autumn (September-November), and winter (December-February). The wavelet analysis was used to determine the periodicity of common respiratory viruses epidemics (25, 26). All statistical analyses were conducted by using R version 3.6.1 (R Foundation for Statistical Computing, Vienna, Austria) (27).

**Virus correlation analysis.** In this study, "virus interaction" refers to when multiple pathogens cocirculate in time and space, which can lead to competitive or cooperative forms of virus–virus (13, 16, 21); "virus interference" describes the situation whereby infection with one virus limits infection and replication of a second virus (20, 28–30). Both virus interaction and virus interference could result in correlations in data patterns. The correlations between respiratory viruses were quantitatively evaluated by employing a simple bivariate cross-correlation analysis based on month detection rate of viruses. The Spearman's rank correlation coefficient was calculated using cor() function in R version 3.6.1 (31). Due to low detection rates for some viruses, before evaluating the correlations, we combined some viruses according to the biological properties (32), in which HPIV-1 and HPIV-3 were combined into HPIV 1/3, HPIV-2 and HPIV-4 were combined into HPIV 2/4, HCoV-229E and HCoV-NL63 were combined into HCoVs-$\alpha$, HCoV-OC43 and HCoV-HKU1 were combined into HCoVs-$\beta$, and RSV-A and RSV-B were combined into RSV. HRVs and EVs were also combined into picoRNA according to the biological characteristics. In addition, IFV-C and HBoV were excluded in the correlation analysis due to the very low detection rates from our data. IFVs refer to the combination of IFV-A and IFV-B. The virus correlation network was conducted by using chordDiagram function in R version 3.6.1 (33).

**Viral interactions estimated by vector autoregressive model.** A vector autoregressive model was applied to identify genuine viral interactions from simple correlations based on time series data from multiple contemporaneous viruses adjusted for confounding factors such as age, gender, and season (34, 35). The model is formulated as:

$$y_t = A_1 y_{t-1} + \cdots + A_p y_{t-p} + B x_t + \varepsilon_t$$

in which $y_t$ indicates k-dimensional endogenous variables, and here refers to the odds of detection ratio monthly of 10 viruses, so k = 10. The $y_t$ values are calculated as the ln of the odds ratio. Given that in some months, the

detection rate is 0 for a virus, the odds ratio was added value 1 before taking the ln-transformation. $x_t$ indicates exogenous variables, including age, gender, and season. $p$ indicates lag phase and lag = 2 was determined here based on the values of AIC. A and B are the parameter matrixes to be evaluated, and $\varepsilon_t$ is random disturbance, assuming to follow a multivariate normal distribution with 0 mean and a 10-by-10 covariance matrix $\Sigma$. Augmented Dickey-Fuller (ADF) test showed that the sequence is stationary, and the impulse response and variance decomposition were performed. The residual correlation matrix associated with $\Sigma$ was used to infer viral interactions. Vector autoregressive model analyses were conducted by R package vars (v 1.5.9). See (34, 35) for further details on the method itself.

**Mathematical models to predict effects of viral interactions on $\beta$ human coronaviruses.** The simulations of the epidemic of $\beta$ human coronaviruses (HCoV-OC43 and HCoV-HKU1) were estimated by using ordinary differential equation (ODE) mathematical models, which assuming a homogenous population without taking into account factors such as migration and death. The simplest SIR model was used. The ODE models are formulated as:

$$\frac{dS}{dt} = -\rho \times I \times S/N$$

$$\frac{dI}{dt} = -\rho \times I \times S/N - \mu \times I$$

$$\frac{dR}{dt} = \mu \times I$$

where $N$ represents the total tested number during the study period, including susceptible, infectious, and recovered participants. $S$ represents the number of susceptible participants, $I$ infectious participants, $R$ recovered participants, $\rho$ daily transmission rate. and $\mu$ the daily cure rate. The specific parameter values are obtained from published literatures and details are provided in the supplemental materials (Table S6) (36). Simulations were performed using ode function in R version 3.6.1 (37).

**Ethics approval.** The ethic was approved by the ethical review committee of the Institute of Pathogen Biology, Chinese Academy of Medical Sciences & Peking Union Medical College (IPB-2014-07). Informed consent was obtained from each patient or their guardians before enrollment.

**Data availability.** The data that support the findings of this study are available from the corresponding author upon reasonable request.

## SUPPLEMENTAL MATERIAL

Supplemental material is available online only.
**SUPPLEMENTAL FILE 1**, PDF file, 0.6 MB.

## ACKNOWLEDGMENTS

We thank the clinicians who contributed to samples collection and transportation.

This study was funded in part by National Major Science & Technology Project for Control and Prevention of Major Infectious Diseases in China (2017ZX10103004), Chinese Academy of Medical Sciences (CAMS) Innovation Fund for Medical Sciences (CIFMS) (2021-I2M-1-038), Fundamental Research Funds for the Central Universities (3332021092), the Nonprofit Central Research Institute Fund of Chinese Academy of Medical Sciences (2019PT310029), Science Fund for Creative Research Groups of the National Natural Science Foundation of China (82221004), the National Natural Science Foundation of China (81930063) and Fondation Mérieux.

Jianwei Wang and Lili Ren conceived and designed experiments. Jianwei Wang, Lili Ren, Xiaojing Dong, and Lulu Zhang analyzed the data and wrote the manuscript. Yan Xiao, Zichun Xiang, Ying Wang, Lan Chen, and Xinming Wang performed the experiments. All authors reviewed the manuscript.

We have no conflicts of interest to declare.

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
