## [Reviewer comments · Microbiology Spectrum]

Microbiology Spectrum

Statistical analysis of common respiratory viruses reveals the binary of virus-virus interaction

Lulu zhang, Yan Xiao, Zichun Xiang, Lan Chen, Ying Wang, Xinming Wang, Xiaojing Dong, Lili Ren, and Jianwei Wang

Corresponding Author(s): Jianwei Wang, Institute of Pathogen Biology, Peking Union Medical College & Chinese Academy of Medical Sciences

Review Timeline:

Submission Date:	January 3, 2023
Editorial Decision:	February 28, 2023
Revision Received:	May 12, 2023
Accepted:	June 9, 2023

Editor: Sen Pei

Reviewer(s): Disclosure of reviewer identity is with reference to reviewer comments included in decision letter(s). The following individuals involved in review of your submission have agreed to reveal their identity: Zhengde Xie (Reviewer #1); Andres Diaz (Reviewer #2)

Transaction Report:

DOI: <https://doi.org/10.1128/spectrum.00019-23>

February 28, 2023

Dr. Jianwei Wang
Institute of Pathogen Biology, Peking Union Medical College & Chinese Academy of Medical Sciences
9 Dong Dan San Tiao, Dongcheng District,
Beijing, Beijing 100730
China

Re: Spectrum00019-23 (**Mathematical analysis of common respiratory viruses reveals the binary of virus-virus interaction and interference**)

Dear Dr. Jianwei Wang:

Thank you for submitting your manuscript to Microbiology Spectrum. Your manuscript has been reviewed by two referees. Based on their comments, we would like to invite a revised manuscript to address their questions. Please find the review comments in the attached files.

Link Not Available

Sincerely,

Sen Pei

Journals Department
Reviewer comments:

Reviewer #1 (Comments for the Author):

Understanding the interaction and interference between epidemics of respiratory viruses is conducive to the formulation of respiratory virus infection control strategies and risk assessment. There are few studies in this field. In this study, the author systematically characterized the interactions between respiratory viruses through mathematical modeling based on the data of multi-pathogen detection. The results showed there are stable interactions among respiratory viruses at population level, which are season-independent. The asynchronous interference between influenza virus and β human coronaviruses was explored, which significantly delayed the peak of β human coronaviruses epidemic during spring and summer.

This is a long-term and large-sample prospective study, with enrolling 14,426 patients suffered from acute respiratory infection (ARI) in Beijing, China during 2005-2015. Therefore, the results of this paper have good reference value, although as the author said, the article still has some limitations.

Minor revision

1. In the discussion, the author should analyze the result: "The interaction between IFVs and HCoV- β was significant in spring and summer, but not in autumn and winter". Why there are different results in different seasons?
2. Another limitation is that children's cases are not included. This should be added to the limitation analysis of the article.

Staff Comments:

Preparing Revision Guidelines

Please return the manuscript within 60 days; if you cannot complete the modification within this time period, please contact me. If you do not wish to modify the manuscript and prefer to submit it to another journal, please notify me of your decision immediately so that the manuscript may be formally withdrawn from consideration by Microbiology Spectrum.

**Mathematical analysis of common respiratory viruses reveals the**
**binary of virus-virus interaction and interference**

Lulu Zhang¹ Ph.D†, Yan Xiao^{1,2} MS†, Zichun Xiang¹ Ph.D, Lan Chen¹ BS, Ying
Wang¹ MS, Xinming Wang¹ MS, Lili Ren^{1,2} Ph.D*, Jianwei Wang^{1,2} Ph.D*

¹ Institute of Pathogen Biology, Chinese Academy of Medical Sciences & Peking
Union Medical College, Beijing 100730, P.R. China

² Key Laboratory of Respiratory Disease Pathogenomics, Chinese Academy of
Medical Sciences and Peking Union Medical College, Beijing 100730, P.R. China

† These authors contributed equally to this work as first authors.

* These authors contributed equally to this work as senior authors.

**Corresponding to:**

Dr. Jianwei Wang

No.9 Dong Dan San Tiao, Dongcheng District, Beijing 100730, P. R. China

Tel/Fax: 86-10-67828516

E-mail: wangjw28@163.com

Dr. Lili Ren

No.9 Dong Dan San Tiao, Dongcheng District, Beijing 100730, P. R. China

Tel/Fax: 86-10-67828516

E-mail: renliliipb@163.com

**ABSTRACT**

Respiratory viruses may interfere with each other and thus affect the epidemic
trend of the virus, but there is little understanding of the interactions between
respiratory viruses at the population level. Here, we conducted a prospective
laboratory-based etiological study by enrolling 14,426 patients suffered from acute
respiratory infection (ARI) in Beijing, China during 2005-2015. All 18 respiratory
viruses were simultaneously tested for each nasal and throat swabs collected from
enrolled patients using molecular tests. The virus interactions were quantitatively
evaluated and confounding factors were also explored by a mathematical model
established in this study. According to the positive and negative correlations, the
respiratory viruses could be divided into two panels. One included influenza viruses A,
B and respiratory syncytial virus, while another panel included human parainfluenza
viruses 1/3, 2/4, adenoviurs, human metapenumovirus, and enterovirus (including
rhinovirus), α and β human coronaviruses, The viruses were positive-correlated in
each panel, negative-correlated among viruses in different panels. The interactions
among respiratory viruses were stable and independent of seasonal distribution. The
asynchronous interference between influenza virus and β human coronaviruses was
explored, which significantly delayed the peak of β human coronaviruses epidemic.
The binary property of the virus interactions indicated transient immunity induced by
one kind of virus would play role on subsequent infection.

**IMPORTANCE:** Systematic quantitative assessment of the interactions between
different respiratory viruses is very important for the prevention of infectious diseases
and the development of vaccine strategies. Our results showed there are stable
interactions among respiratory viruses at population level, which are
season-independent. Respiratory viruses could be divided into two panels. One
included influenza virus and respiratory syncytial virus, while another included the
others. It showed negative interactions between the two panels. The asynchronous
interference between influenza virus and β human coronaviruses significantly delayed
the peak of β human coronaviruses epidemic. The binary property of the viruses
indicated transient immunity induced by one kind of virus would play role on
subsequent infection, which provides important data for the development of epidemic
surveillance strategies.

**KEYWORDS:** acute respiratory infection, population level, viral interactions,
asynchronous interference, transient immunity, influenza virus, β human
coronaviruses, epidemic surveillance

**Running Title:** The binary of virus interaction and interference.

INTRODUCTION

Acute respiratory infection (ARI) is the major cause of morbidity and mortality
worldwide. Each person would be attacked 2-3 episodes of ARI each year (1-3) and a
quarter of the population would need primary care (4). Respiratory viruses are the
most common agents of ARI, including influenza viruses (IFVs), respiratory syncytial
virus (RSV), human coronaviruses (HCoVs, including 229E, OC43, NL63 and
HKU1), parainfluenza virus (HPIV) 1-4, enterovirus (EVs)/human rhinovirus (HRVs),
adenovirus (Adv), human metapneumovirus (hMPV), human bocavirus (HBoV) and
etc. (5, 6). These viruses have different epidemic patterns, population susceptibility,
and mechanisms of infection (6-9). The ecology of respiratory tract depends on the
dynamic regulation among viruses, bacteria and host. At present, the viruses-host,
bacteria-host and viruses-bacteria interactions have been widely studied (10-13), but
the research on virus-virus interactions is relatively little. Elucidating this gap in
knowledge will help prevent and treat respiratory infectious diseases.

Viral interference has long been recognized and reported. During the 2009
pandemic of an emerging IFV-A, when data from several European countries
indicated that the annual autumn HRVs epidemic interrupted and delayed
transmission of the emerging IFVs (13-15). Studies from the United Kingdom
provided the first systematic large-scale presentation of respiratory virus interactions
(16). Clinical studies and experimental animal models also provide initial evidence of
viral interactions at the host level (17). The mechanism behind virus-virus interactions
is not clear. Adaptive immunity is one of the possible explanations (17, 18).

Cross-immunity caused by infected with one virus can alter the epidemic dynamics of
another virus (19). Other possible explanations include competition for resources and
other biological processes (20).

Although sporadic studies have shown viral interactions, they are limited. The
reason is that the limited co-epidemic data involving all the common respiratory
viruses restrict us the understanding on interactions among the respiratory viruses
over a long period. And it is difficult to explain whether the phenomenon is
etiology-driven or host-variable driven even data are available.

In this study, we conducted a prospective laboratory-based respiratory viruses
etiological study by enrolling patients suffered from ARI in Beijing, China during
2005-2015. Each enrolled patient was simultaneously tested for 18 respiratory viruses.
We used statistical methods and models to elucidating the interaction characteristics
of respiratory viruses and the effect of host confounding factors. In addition, taking
human coronaviruses as example, we found the effect of viral interactions on virus
prevalence, which provides important data reference for the development of epidemic
surveillance strategies for respiratory infectious diseases.

**RESULTS**

**Detections of respiratory viruses**

A total of 14,426 adult outpatients with ARI were enrolled, in which 963 (6.68%)
aged ≥ 65 years, and 954 (6.61%) had an underlying disease during visiting. The most
common underlying disease was hypertension (n=306, 2.12%), followed by chronic

liver, heart or renal diseases (n=196, 1.36%), diabetes (n=144, 1.00%), cancer (n=133,
0.92%) and chronic lung diseases (n=83, 0.58%). The elderly patients had the most
underlying diseases (n=267/963, 27.73%, $P<0.001$) among all age groups.
Self-administrated antibiotics usage before visiting was reported in 2,082 (14.43%)
patients. Antivirals usage before visiting was reported in 26 (<1%) (**Table S1**).

At least one virus was detected in 5,585 patients (38.71%), in which 5,243
(36.34%) were single-detected and 342 (2.37%) multiple-detected. The most frequent
single-detected virus was IFV-A (n=2,399, 16.63%), followed by IFV-B (n=814,
5.64%), HRVs (n=798, 5.53%), EVs (n=355, 2.46%), and Adv (n=151, 1.05%). The
detection rates of other 13 respiratory viruses were less than 1% (**Table S2**). HRVs
(n=173, 50.58%), IFVA (n=165, 48.25%) and IFVB (n=73, 21.35%) were the most
frequently detected viruses in the cases with multiple pathogens co-detected. IFV-A
with HRVs (n=83, 24.27%) was the most common dual-detected viruses (**Table S3**).

The overall detection rates of respiratory viruses varied monthly with a range of
0%-86.67% (**Fig. 1A**). According to the mainly peaked season, the 18 respiratory
viruses were subjectively grouped into four groups (**Fig. 1B**). Group I included hMPV,
Adv and HPIV-4, mainly detected in spring. Group II included HPIVs 1-3, EVs,
HCoV-OC43 and HCoV-HKU1, mainly detected in summer. Group III included
HRVs, HBoV, HCoV-229E and HCoV-NL63, which were in slightly higher detection
rates in autumn. Group IV included IFVs and RSV, mainly detected in winter. The
prevalence of IFV-A, Adv, EVs and hMPV showed significant periodicity ($P<0.05$)

when analyzed by using wavelet method, while IFVB, RSV, HPIVs, HRVs and
HCoVs showed no significant epidemic periodicity ($P>0.05$) (Fig. 1C, Fig. S1).

The detection rate of respiratory virus showed different among age groups
($P=0.023$), with the highest rate in 14-24 years old (40.69%) and lowest in ≥ 65 years
old (37.69%) (Table S2). HRVs, EVs and Adv were more frequently detected in
14-24 years, IFVs in 45-64 years, while RSV, HPIVs and HCoVs in elderly adults
($P<0.05$) (Fig. 2, Table S2).

Interactions between respiratory viruses

To test whether there exist interactions between respiratory viruses, we analyzed
the correlations between respiratory viruses by using Spearman's rank correlation test.
The data showed seven negative correlations and seven positive correlations among
these viruses (Fig. 3A). The positive correlations were found between RSV with
IFV-B ($r=0.56$, $P<0.001$), HPIV 1/3 with HPIV 2/4 ($r=0.35$, $P<0.001$), HCoVs- β with
HPIV 2/4 ($r=0.31$, $P=0.001$), Adv with HCoVs- α ($r=0.21$, $P=0.021$), Adv with hMPV
($r=0.30$, $P=0.001$), HPIV 2/4 with picoRNA ($r=0.22$, $P=0.015$) and HPIV 1/3 with
picoRNA ($r=0.24$, $P=0.009$). Negative correlations were observed between IFV-A
with picoRNA ($r=-0.23$, $P=0.012$), IFV-B with picoRNA ($r=-0.33$, $P<0.001$), RSV
with picoRNA ($r=-0.29$, $P=0.001$), HPIV 2/4 with IFV-A ($r=-0.23$, $P=0.013$), HPIV
2/4 with RSV ($r=-0.25$, $P=0.006$), HPIV 1/3 with IFV-A ($r=-0.22$, $P=0.016$) and HPIV
1/3 with RSV ($r=-0.20$, $P=0.031$) (Fig. 3A, Fig. S2).

Based on the combinations of positive and negative correlated viruses, the
respiratory viruses could be further divided into two panels. One panel included

IFV-A, IFV-B and RSV, while another panel included HPIV 1/3, HPIV 2/4, HCoV α ,
HCoV β , Adv, hMPV, and picoRNA. Within each panel, all the viruses showed
positive correlations, while viruses in different panel showed negative correlations
(Fig. 3B). As exemplified, IFV-A, IFV-B and RSV, the main pathogens of respiratory
infections in winter, showed negative correlations with all the other tested viruses in
our study.

**Influence of confounding factors on virus interactions**

We further evaluated the stability of the observed interactions between viruses
across ten years. It showed that the virus interactions fluctuated between -0.50 and
0.50 without zero during 2005 to 2015, indicating the virus interactions were stable
(Fig. 4A).

A new mathematical model was developed in this study to explore the factors
influencing virus interactions, including gender, age and seasons. It showed that
seasonality contributed no effect on virus interactions (Fig. 4B, Table S4 and Table
S5). Based on the effect weights, age was found to be the most important factor in the
interactions between HPIV 1/3 and HPIV 2/4 (0.15 vs. 0.10), HPIV 2/4 and HCoV β
(0.10 vs. 0.18), and IFV-B and picoRNA (0.02 vs. 0.02). Similarly, the interaction of
HPIV 1/3 with IFV-A (0.06 vs. 0.05) is mainly influenced by gender (Fig. 4B, Table
S4 and Table S5). Such findings suggested that age and gender, the demographic
factors should be considered intensively when decided the correlations among
respiratory viruses.

**Delaying effect of viral interactions on the epidemic of β human coronaviruses**

During the study period, the epidemic of HCoV β (HCoV-OC43 and
HCoV-HKU1) were at a relative low level and mainly peaked in summer. Since we
have obtained the stable negative correlation of IFVs with the common respiratory
viruses except RSV. This model would help us to find viral factors to influence the
epidemic activity of HCoV β . A significant asynchronous interference between IFVs
and HCoV β was identified in spring and summer (spring: $r=-0.04$, $P=0.029$;
summer: $r=-0.04$, $P=0.009$), but not in autumn and winter (**Fig. 5A and B**). We used
Susceptible-Infectious-Recovered (SIR) model to simulate the influence of
asynchronous interference on the epidemic of HCoV β . It showed that the peaking
epidemic time of HCoV β would be delayed from original 120 days to 130 days in
spring and 135 days in summer if asynchronous interference was involved (**Fig. 5C,**
**5D**). This model was then used to simulate the effect of viral interference on the
epidemic of β human coronaviruses with different basic transmission rates ($R_0=2.70$,
5.50 and 8.00). It showed that the impact of asynchronous interference of IFVs on the
epidemic activity of β human coronaviruses decreased with the increasing of R_0 (**Fig.**
**5E and F**). If R_0 reaches to 8.00, the effect of IFVs asynchronous interference hardly
affects the peak time of β human coronaviruses.

**DISCUSSION**

In this study, we characterized the distribution and interactions of respiratory
viruses from 2005 to 2015, based on a laboratory-based screening on patients suffered
from ARI. We found there exists stable correlations between respiratory viruses across
the studied ten years. IFVs combined with RSV, showed negative correlations with
other common respiratory viruses. IFVs showed asynchronous interference with β
human coronaviruses by delaying the duration of the epidemic peak.

Virus interaction is an important interfering factor affecting the epidemic
magnitude, incidence and peak of respiratory pathogens (13, 21, 22). Viral
transmissibility, human exposure history, and the cross immunity induced by the
infected viruses were the potential mechanisms on virus interactions (15, 21, 22).
Jenner first reported in 1804 that herpes infections may prevent the development of
vaccinia lesions, which was later named viral interference (23). Till now, the
interactions among viruses and the effects were still not well defined. Among IFVs,
RSV and HPIVs, viruses with high epidemic activities, the neutral or competitive
effects among the viruses were still in debated considering the population distribution,
individual incidence, and laboratory investigations (13, 22). One of the major reasons
is the lack of epidemiological data, the absent of mathematical models and integrated
analysis.

Our study integrated self-designed mathematics models, ecological and
demographic factors and laboratory-based viral screening data obtained across ten
207 years. We found all the tested viruses showed a long-term and stable process,
independent of season, while age and gender are the major confounding factors on

virus interactions. This partially explain why the special epidemic patterns of viruses
are at different age stages and why some viruses are more common in men or women.
It has been suggested that different gender or age group have different immune
responses to viral infections (24). However, the interactions among different viruses
can not explain the seasonality of different viruses. Such findings emphasized a
limited role of ecological factors in virus interactions and virus-host interactions.

The asynchronous interference between IFV-A and HRVs found in this study has
also been reported in other studies (16, 17), although studies were conducted in
different populations, suggesting that transient immunity from IFV-A infection is
effective against HRVs. In addition, there were significant negative interferences
between IFVs, RSV, and other viruses, indicating that immunity to IFVs and RSV
may be effective against other viruses as well. These findings will be important for the
development of vaccine strategies.

Although asynchronous interference with IFVs can significantly delay the
epidemic peak of HCoVs- β during spring and summer, it has limited effect on viruses
with high transmission rate. These results suggested that we should be more vigilant
against co-infection of high transmissible strains with IFVs. Although our study used
the simplest SIR model, these results provide important reference data for epidemic
prevention and control in the post-COVID-19 era, which needs to pay attention to
co-infection of SARS-CoV-2 and IFVs (25, 26).

There are some **limitations** in our study. One is that the study is a hospital-based
investigation, while our findings might be further confirmed by involved much more

data from the community. Another is that our study was conducted at Beijing, which
located in the north of China. Despite some of the recruited cases were from the south
of China, the generality of virus interactions need to be evaluated intensively in the
future by involving more data from other geographical regions.

In summary, our study systematically characterized the interactions between
respiratory viruses through mathematical modeling based on the data of
multi-pathogen detection in large samples, and revealed the binary of viral
interactions. These results will improve our understanding on the epidemic of
respiratory viruses in natural. Further understanding of the biological mechanisms
underlying viral interactions could help improve disease prediction and evaluation
policies in the future.

**MATERIALS AND METHODS**

**Study design**

From May 1, 2005 through April 30, 2015, adults suffered from ARI, aged no
less than 14 years were enrolled at the Fever Outpatient Clinic Department (FOCD) in
Peking Union Medical College Hospital, Beijing, China. The first 5-10 patients met
the criteria of ARI suspected viral infections were enrolled in each week during the
study period. The recruitment criteria include: (1) acute onset; (2) with fever (body
temperature $\geq 38^{\circ}\text{C}$); (3) with normal or low leukocyte count (a white blood cell count
$\leq 10,000/\text{ml}$); (4) symptoms or signs of respiratory tract infections. Patients were
excluded if they had been hospitalized one week previously, aged less than 14 years,

or had been enrolled in the study within the previous 28 days. Demographic and
clinical data were collected from each patient by using a standardized case reporting
form. For each enrolled patient, nose and throat swabs were collected and the two
swabs were pooled in one tube containing virus transport medium (VTM; Copan,
Brescia, Italy).

**Respiratory viruses screening**

Total nucleic acid (DNA and RNA) was directly extracted from each specimen
by using the NucliSens easyMAG apparatus (bioMérieux, Marcy l'Etoile, France),
according to the manufacturer's instructions. The presence of IFVs A, B and C,
HPIVs 1–4, RSV A and B, Adv, HBoV, hMPV, EVs (including HRVs), and
HCoV-229E, HCoV-OC43, HCoV-NL63 and HCoV-HKU1 were screened by using
reverse-transcriptase polymerase chain reaction (RT-PCR) and PCR method. Detailed
information on primers and laboratory procedures has been summarized previously
(6).

**Statistical analysis**

The detection rate of virus was calculated by dividing the number of positive
cases by the total tested number. The proportions of positive detections were
compared by using Chi-square or Fisher's exact tests. A two-sided *P* value of <0.05
was considered statistically significant. The monthly distribution was scaled from 0 to
1 of each virus, according to the percentile rank. Four seasons were determined as
spring (March-May), summer (June-August), autumn (September-November), and
winter (December-February). The wavelet analysis was used to determine the
periodicity of common respiratory viruses epidemics (27, 28). All statistical analyses

were conducted by using R version 3.6.1 (R Foundation for Statistical Computing,
Vienna, Austria) (29).

**Virus interactions analysis**

The correlations between respiratory viruses were quantitatively evaluated by
employing a simple bivariate non-parametric cross-correlation analysis. The
Spearman's rank correlation coefficient was calculated using cor function in R version
3.6.1 (30). To avoid bias caused by low detection rate, we combined the positive
detections of the virus with (<1%) according to the biological properties (31), in
which HPIV-1 and HPIV-3 were combined into HPIV 1/3, HPIV-2 and HPIV-4 were
combined into HPIV 2/4, HCoV-229E and HCoV-NL63 were combined into
HCoVs- α , HCoV-OC43 and HCoV-HKU1 were combined into HCoVs- β , RSV-A and
RSV-B were combined into RSV. HRVs and EVs were also combined into picoRNA
according to the biological characteristics. In addition, IFV-C and HBoV were
excluded in the correlation analysis due to the very low detection rate. IFVs were then
indicated the combination of IFV-A and IFV-B. The virus interactions network was
conducted by using chordDiagram function in R version 3.6.1 (32).

**Mathematical models to evaluate effect of driver factors on virus interactions**

A regression model was established to evaluate the effect of drivers on virus
interactions. The known confounding factors related to viral detection rate were
involved, including age, gender and seasons. Six variables were included in the model,
including patient age (group1[14-44y], group2 [45-64y]), patient gender of male,
seasons, as well as two interactive items, gender.male \times age. group1, gender.male \times

age.group2. The stepwise regression was used to characterize the effects of each
 factor. Thus, each virus model (HPIV 1/3, HPIV 2/4, IFV-A, IFV-B, HCoV-s- α ,
 HCoV-s- β , RSV, Adv, hMPV, picoRNA) may generate six correlation coefficients
 ($\beta_1 \dots \beta_6$) for six variables and one constant coefficients β_0 . In addition, we calculated
 the mean value of detection rate for each virus (mean (Y)). The weight of the effect of
 each factor is expressed by $\beta_i / \text{mean}(Y)$. The model is formulated as:

$$304 \quad \text{Logit}(Y) = \beta_0 + \beta_1 \times \text{gender.male} + \beta_2 \times \text{age.group1} + \beta_3 \times \text{age.group2} \\
 + \beta_4 \times \text{gender.male} \times \text{age.group1} + \beta_5 \times \text{gender.male} \times \text{age.group2} + \beta_6 \times \text{season} \quad (1)$$

$$305 \quad \text{Weight}(f_i) = \beta_i / \text{mean}(Y), i = 1, 2, \dots, 6 \quad (2)$$

with Y indicating the prevalence rate, $\beta_1 \dots \beta_6$ the regression coefficients of six
 variables, and f_i , the variables influencing the viral detection rate. The weight of the
 same factor for different viruses were compared to clarify whether it is the main factor
 related to the detection rate. Regression analyses were conducted by using the SPSS
 (version 19.0).

**Mathematical models to predict effects of viral interactions on β human**
 **coronaviruses**

The simulations of the epidemic of β human coronaviruses (HCoV-OC43 and
 HCoV-HKU1) were estimated by using ordinary differential equation (ODE)
 mathematical models, which assuming a homogenous population without taking into
 account factors such as migration and death. The simplest SIR model was used. The
 ODE models are formulated as:

$$\frac{dS}{dt} = -\rho \times I \times S / N \quad (3)$$

$$\frac{dI}{dt} = -\rho \times I \times S / N - \mu \times I \quad (4)$$

$$\frac{dR}{dt} = \mu \times I \quad (5)$$

where N represents the total tested number during the study period, including
susceptible, infectious and recovered participants. S represents the number of
susceptible participants, I for infectious participants and R for recovered participants,
ρ , the daily transmission rate and μ , the daily cure rate. Details of the parameters are
provided in the supplementary materials (Table S6) (33). Simulations were performed
using ode function in R version 3.6.1 (34).

**Ethics approval**

The ethic was approved by the ethical review committee of the Institute of
Pathogen Biology, Chinese Academy of Medical Sciences & Peking Union Medical
College (IPB-2014-07). Informed consent was obtained from each patient or their
guardians before enrollment.

**Data availability**

The data that support the findings of this study are available from the
corresponding author upon reasonable request.

**ACKNOWLEDGMENTS**

**We would like to thank Professor Xiaojing Dong (Santa Clara University)**
**for her support in data analysis and the clinicians who contributed to samples**
**collection and transportation.**

**Funding: This study was funded in part by National Major Science &**
**Technology Project for Control and Prevention of Major Infectious Diseases in**
**China (2017ZX10103004), Chinese Academy of Medical Sciences (CAMS)**
**Innovation Fund for Medical Sciences (CIFMS) (2021-I2M-1-038), Fundamental**
**Research Funds for the Central Universities (3332021092), the Nonprofit Central**
**Research Institute Fund of Chinese Academy of Medical Sciences**
**(2019PT310029), Science Fund for Creative Research Groups of the National**
**Natural Science Foundation of China (82221004), the National Natural Science**
**Foundation of China (81930063) and Fondation Mérieux.**
**Author Contributors: Jianwei Wang and Lili Ren conceived and designed**
**experiments. Yan Xiao, Zichun Xiang, Ying Wang, Lan Chen and Xinming Wang**
**performed the experiments. Jianwei Wang, Lili Ren and Lulu Zhang analyzed**
**the data and wrote the manuscript. All authors reviewed the manuscript.**
**Declaration of interests: All authors declare no competing interests.**

**REFERENCES**

- 1. GBD 2017 Disease and Injury Incidence and Prevalence Collaborators. 2018.
Global, regional, and national incidence, prevalence, and years lived with
disability for 354 diseases and injuries for 195 countries and territories,
1990-2017: a systematic analysis for the Global Burden of Disease Study 2017.
Lancet 392:1789-1858.
- 2. GBD 2019 LRI Collaborators. 2022. Age-sex differences in the global burden of

- lower respiratory infections and risk factors, 1990-2019: results from the Global
Burden of Disease Study 2019. *Lancet Infect Dis* Aug 11.
- 3. GBD 2019 Risk Factors Collaborators. 2020. Global burden of 87 risk factors in
204 countries and territories, 1990-2019: a systematic analysis for the Global
Burden of Disease Study 2019. *Lancet* 396: 1223-1249.
- 4. Li Y, Wang X, Blau D, Caballero M, Feikin D, Gill C, Madhi S, Omer S, Simões
E, Campbell H, Pariente A, Bardach D, Bassat Q, Casalegno J, Chakhunashvili G,
Crawford N, Danilenko D, Do L, Echavarria M, Gentile A, Gordon A, Heikkinen
370 T, Huang Q, Jullien S, Krishnan A, Lopez E, Markić J, Mira-Iglesias A, Moore H,
Moyes J, Mwananyanda L, Nokes D, Noordeen F, Obodai E, Palani N, Romero C,
Salimi V, Satav A, Seo E, Shchomak Z, Singleton R, Stolyarov K, Stoszek S, von
G, Wurzel D, Yoshida L, Yung C, Zar H. 2022. Global, regional, and national
disease burden estimates of acute lower respiratory infections due to respiratory
syncytial virus in children younger than 5 years in 2019: a systematic analysis.
*Lancet* 399: 2047-2064.
- 5. Organization WH. 2013. Research needs for the battle against respiratory viruses
(brave) Geneva, Switzerland: WHO Press.
- 6. Ren L, Gonzalez R, Wang Z, Xiang Z, Wang Y, Zhou H, Li J, Xiao Y, Yang Q,
Zhang J, Chen L, Wang W, Li Y, Li T, Meng X, Zhang Y, Vernet G,
Paranhos-Baccalà G, Chen J., Jin Q, Wang J. 2009. Prevalence of human
respiratory viruses in adults with acute respiratory tract infections in Beijing,
2005-2007. *Clin Microbiol Infect* 15: 1146-1153.

- 7. Yu J, Xie Z, Zhang T, Lu Y, Fan H, Yang D, Bénet T, Vanhems P, Shen K,
Huang F, Han J, Li T, Gao Z, Ren L, Wang J. 2018. Comparison of the
prevalence of respiratory viruses in patients with acute respiratory infections at
different hospital settings in North China, 2012-2015. *BMC Infect Dis* 18: 72.
- 8. Jain S, Self W, Wunderink R, Fakhran S, Balk R, Bramley A, Reed C, Grijalva C,
Anderson E, Courtney D, Chappell J, Qi C, Hart E, Carroll F, Trabue C,
Donnelly H, Williams D, Zhu Y, Arnold S, Ampofo K, Waterer G, Levine M,
Lindstrom S, Winchell J, Katz J, Erdman D, Schneider E, Hicks L, McCullers J,
Pavia A, Edwards K, Finelli L, CDC EPIC Study Team. 2015.
Community-Acquired Pneumonia Requiring Hospitalization among U.S. Adults.
*N Engl J Med* 373: 415-427.
- 9. Li Z, Zhang H, Ren L, Lu Q, Ren X, Zhang C, Wang Y, Lin S, Zhang X, Li J,
Zhao S, Yi Z, Chen X, Yang Z, Meng L, Wang X, Liu Y, Wang X, Cui A, Lai S,
Jiang T, Yuan Y, Shi L, Liu M, Zhu Y, Zhang A, Zhang Z, Yang Y, Ward M,
Feng L, Jing H, Huang L, Xu W, Chen Y, Wu J, Yuan Z, Li M, Wang Y, Wang L,
Fang L, Liu W, Hay S, Gao G, Yang, Chinese Centers for Disease Control and
Prevention (CDC) Etiology of Respiratory Infection Surveillance Study Team.
2021. Etiological and epidemiological features of acute respiratory infections in
China. ***Nat Commun* 12: 5026.**
- 10. Chambers B, Heaton B, Rausch K, Dumm R, Hamilton J, Cherry S, Heaton N.
2019. DNA mismatch repair is required for the host innate response and controls
cellular fate after influenza virus infection. ***Nat Microbiol* 4: 1964-1977.**

- 11. Palmer J, Foster K. 2022. Bacterial species rarely work together. *Science* 376:
581-582.
- 12. Rowe H, Meliopoulos V, Iverson A, Bomme P, Schultz-Cherry S, Rosch J. 2019.
Direct interactions with influenza promote bacterial adherence during respiratory
infections. *Nat Microbiol* 4: 1328-1336.
- 13. Bosch A, Biesbroek G, Trzcinski K, Sanders E, Bogaert D. 2013. Viral and
bacterial interactions in the upper respiratory tract. ***PLoS Pathog* 9(1): e1003057.**
- 14. Casalegno J, Ottmann M, Duchamp M, Escuret V, Billaud G, Frobert E, Morfin F,
Lina B. 2010. Rhinoviruses delayed the circulation of the pandemic influenza A
(H1N1) 2009 virus in France. *Clin Microbiol Infect* 16: 326–329.
- 15. Long Q, Tang X, Shi Q, Li Q, Deng H, Yuan J, Hu J, Xu W, Zhang Y, Lv F, Su
417 K, Zhang F, Gong J, Wu B, Liu X, Li J, Qiu J, Chen J, Huang A. 2020. Clinical
and immunological assessment of asymptomatic SARS-CoV-2 infections. *Nat*
*Med* 26: 1200-1204.
- 16. Nickbakhsh S, Mair C, Matthews L, Reeve R, Johnson P, Thorburn F, von W,
Reynolds A, McMenamin J, Gunson R, Murcia P. 2019. Virus-virus interactions
impact the population dynamics of influenza and the common cold. *Proc Natl*
*Acad Sci* 116(52): 27142–27150.
- 17. Wu A, Mihaylova V, Landry M, Foxman E. 2020. Interference between
rhinovirus and influenza A virus: a clinical data analysis and experimental
infection study. *Lancet Microbe* 1: e254-e262.
- 18. Opatowski L, Baguelin M, Eggo R. 2018. Influenza interaction with cocirculating

- pathogens and its impact on surveillance, pathogenesis, and epidemic profile: A
key role for mathematical modelling. *PLoS Pathog* 14: e1006770.
- 19. Bhattacharyya S, Gesteland P, Korgenski K, Bjørnstad O, Adler F. 2015.
Cross-immunity between strains explains the dynamical pattern of
paramyxoviruses. *Proc Natl Acad Sci* 12: 13396-1400.
- 20. Chan K, Carolan L, Korenkov D, Druce J, McCaw J, Reading P, Barr I, Laurie K.
2018. Investigating Viral Interference Between Influenza A Virus and Human
Respiratory Syncytial Virus in a Ferret Model of Infection. *J Infect Dis* 218:
406-417.
- 21. Shaman J, Galanti M, 2020. Will SARS-CoV-2 become endemic? *Science* 370,
527-529.
- 22. Karppinen S, Toivonen L, Schuez-Havupalo L, Waris M, Peltola V. 2016.
Interference between respiratory syncytial virus and rhinovirus in respiratory tract
infections in children. *Clin Microbiol Infect* 22(2): 208.e1-208.
- 23. Jenner E. 1804 on the Effects of Cutaneous Eruptions. *Med Phys J* 12(66):
97-102.
- 24. Adland E, Millar J, Bengu N, Muenchhoff M, Fillis R, Sprenger K, Ntlantsana V,
Roider J, Vieira V, Govender K, Adamson J, Nxele N, Ochsenbauer C, Kappes J,
Mori L, van L, Graza Y, Chinniah K, Kapongo C, Bhoola R, Krishna M,
Matthews P, Poderos R, Lluch M, Puertas M, Prado J, McKerrow N, Archary M,
Ndung'u T, Groll A, Jooste P, Martinez-Picado J, Altfeld M, Goulder P. 2020.
Sex-specific innate immune selection of HIV-1 in utero is associated with

- increased female susceptibility to infection. *Nat Commun* 11: 1767.
- 25. Koutsakos M, Wheatley A, Laurie K, Kent S, Rockman S. 2021 Influenza lineage
extinction during the COVID-19 pandemic? *Nat Rev Microbiol* 19: 741-742.
- 26. Laurie K, Rockman S. 2021. Which influenza viruses will emerge following the
SARS-CoV-2 pandemic? *Influenza Other Respir Viruses* 15: 573-576.
- 27. Chigusa S, Moroi T, Shoji Y. 2017. State-of-the-Art Calculation of the Decay
Rate of Electroweak Vacuum in the Standard Model. *Phys Rev Lett* 119: 211801.
- 28. Cheng X, Tan Y, He M, Lam T, Lu X, Viboud C, He J, Zhang S, Lu J, Wu C,
Fang S, Wang X, Xie X, Ma H, Nelson M, Kung H, Holmes E, Cheng J. 2013.
Epidemiological dynamics and phylogeography of influenza virus in southern
China. *J Infect Dis* 207: 106-114.
- 29. R Core Team. 2013. *R: A Language and environment for statistical computing*
(R Foundation for Statistical Computing, Vienna, 2013).
- 30. Astivia O, Zumbo B. 2017. Population models and simulation methods: The case
of the Spearman rank correlation. *Br J Math Stat Psychol* 70: 347-367.
- 31. Goka E, Vallely P, Mutton K, Klapper P. 2014. Single and multiple respiratory
virus infections and severity of respiratory disease: a systematic review. *Paediatr*
*Respir Rev* 15: 363–370.
- 32. Gu Z. 2014. Circlize implements and enhances circular visualization in R.
*Bioinformatics* 30(19): 2811-2812.
- 33. Kissler S, Tedijanto C, Goldstein E, Grad Y, Lipsitch M. 2020. Projecting the
transmission dynamics of SARS-CoV-2 through the postpandemic period.

Science 368: 860-868.

34. Li Q, Guan X, Wu P, Wang X, Zhou L, Tong Y, Ren R, Leung K, Lau E, Wong J,
Xing X, Xiang N, Wu Y, Li C, Chen Q, Li D, Liu T, Zhao J, Liu M, Tu W, Chen
C, Jin L, Yang R, Wang Q, Zhou S, Wang R, Liu H, Luo Y, Liu Y, Shao G, Li H,
Tao Z, Yang Y, Deng Z, Liu B, Ma Z, Zhang Y, Shi G, Lam T, Wu J, Gao G,
Cowling B, Yang B, Leung G, Feng Z. 2020. Early transmission dynamics in
Wuhan, China, of novel coronavirus-infected pneumonia. N Engl J Med 382:
1199-1207.

**FIGURE LEGENDS**

**Fig. 1. Temporal distribution of viral pathogens in adults with acute respiratory**
**infections in Beijing, China, 2005-2015. A.** Overall monthly prevalence of
respiratory viruses; **B.** Thermodynamic diagram of average monthly prevalence,
scaled from 0 to 1 according to percentile rank, by viruses; **C.** Thermodynamic
diagram of yearly prevalence, scaled from 0 to 1 according to percentile rank, by
viruses. Abbreviations: IFVs (A, B, C) = influenza virus (type A, B, C); HRVs =
human rhinoviruses; HPIVs (1, 2, 3, 4) = human parainfluenza viruses (type 1, 2, 3, 4);
EVs = enteroviruses; Adv = adenoviruses; HCoV (NL63, HKU1, OC43, 229E) =
human coronaviruses (type NL63, HKU1, OC43, 229E); RSV (A, B) = respiratory
syncytial virus (subgroup A, B); hMPV = human metapneumovirus; HBoV = human
bocaviruses.

**Fig. 2. Age distribution of viral pathogens in adults with acute respiratory**
**infections in Beijing, China, 2005-2015. A.** No. of detections, by viruses; **B.**
Proportions of detections, by viruses; **C.** Proportions of detections, by viruses and
illness onset season. Abbreviations: IFVs (A, B, C) = influenza virus (type A, B, C);
HRVs = human rhinoviruses; HPIVs (1, 2, 3, 4) = human parainfluenza viruses (type
1, 2, 3, 4); EVs = enteroviruses; Adv = adenoviruses; HCoV (NL63, HKU1, OC43,
229E) = human coronaviruses (type NL63, HKU1, OC43, 229E); RSV (A, B) =
respiratory syncytial virus (subgroup A, B).

**Fig. 3. Viral interactions. A.** Correlations among respiratory viruses at the population
level. The heatmap represents correlations between common respiratory viruses, in
which blue squares represent positive correlations and red squares represent negative
correlations. The larger the square, the darker the color, and the higher the correlation.
Correlations with P values of statistical tests less than 0.05 were marked with an
asterisk. **B.** Virus interaction network of common respiratory viruses. The blue and
green strips represent positive interactions, and the orange and rose strips represent
negative interactions. The thickness of the strip represents the size of the interaction.
The ability of each virus to make connections with other viruses is marked in the
middle circle. All these correlations were statistically significant (Spearman's rank
correlation test, $P < 0.05$). Abbreviations: IFVs (A, B, C) = influenza virus (type A, B,
C); HPIV 1/3 = human parainfluenza viruses (type 1, 3); HPIV 2/4 = human
parainfluenza viruses (type 2, 4); HCoV- α = human coronaviruses (type NL63, 229E);
HCoV- β = human coronaviruses (type OC43, HKU1); hMPV = human
metapneumovirus; Adv = adenoviruses; RSV (A, B) = respiratory syncytial virus
(subgroup A, B); picoRNA = picornaviridae (including human rhinoviruses and
enteroviruses).

**Fig. 4. Factors affecting viral interactions. A.** The temporal density map of virus
interactions from 2005 to 2015. **B.** Effect weight of host factors and seasonal factors
on the prevalence of respiratory virus.

**Fig. 5. The Dynamic model simulates the effect of viral interactions on β human**
**coronaviruses prevalence. A.** The prevalence of IFVs showed no correlations with
HCoV β s (HCoV-OC43 and HCoV-HKU1) by using Spearman's rank correlation test. 529 **B.** The interaction between IFVs and HCoV β s was significant in spring and summer,
but not in autumn and winter. **C.** SIR (Susceptible-Infectious-Recovered) models
simulate transmission dynamic of HCoV β s. **D.** The model predicts the impact of
IFVs asynchronous interference on HCoV β s epidemic. **E.** SIR models simulate the
transmission dynamic of β human coronaviruses with different transmission rates
($R_0=2.7, 5.5$ and 8.0). **F.** The model predicts the impact of IFVs asynchronous
interference on β human coronaviruses ($R_0=2.7, 5.5$ and 8.0) epidemic.

A

Density heatmap of Interaction

**B****gender.male****age.group1****age.group2****gender.male*age.group1****gender.male*age.group2****season**
A Interaction between IFV-A and HCoVVs- β **B** Interaction between IFV-A and HCoVVs- β **C** HCoVVs- β **D** HCoVVs- β **E** beta coronavirus (R0=2.7)
beta coronavirus (R0=5.5)

beta coronavirus (R0=8)

**F** beta coronavirus (R0=2.7)
beta coronavirus (R0=5.5)

beta coronavirus (R0=8)

In the manuscript “Mathematical analysis of common respiratory viruses reveals the binary of virus-interactions and interference” Zhang et al., report a fascinating interaction of human respiratory viruses based on a prospective laboratory-based study with 14426 patients with acute respiratory infection (ARI) in Beijing between 2005 and 2015. Most conclusions are supported by the authors’ findings. However, several clarifications regarding the methods and interpretation must be addressed to make the paper clearer and stronger. The majority of the manuscript is well written, but a final English proofread is recommended to identify minor grammatical errors prior publication.

General comments:

1. I believe the proposed models are more epidemiological or statistical than mathematical. Consider adjusting the title to “Statistical analysis of...”.
2. It would be useful to better define virus “interaction”, “interference”, and “correlation” in the methods sections, and use these terms consistently through the paper to avoid confusion. Even in the title the authors use “interaction and interference” without giving clarity of its definition for the purpose of the study.
3. “Interaction” in statistics or epidemiology differs from what the authors call virus “interactions” or correlations. It seems that these terms (interactions, interference and correlations) are used across the manuscript synonymously but it is not clear for the reader what comes from the interaction terms in model 1 (line 304), the Spearman’s rank test (line 281), and what yields the density heat map of interactions shown in Figure 4.
4. A better explanation of the models used to estimate interactions and correlation may help the reader to follow smoothly the study results.

Specific comments:

Methods sections:

Line 249: clarify if the remaining patients (after the 5-10 recruited) were not included in the study.

Line 268: Detection rate. Stay consistently through the paper and do not use “prevalence” as it is not a prevalence study. Legend for figure 1 (line 485) should not read “Monthly prevalence”

Line 278: Is a correlation analysis hence a correlation coefficient is obtained. Not an interaction analysis. Change the subtitle of this section.

Given the logistic regression model used, the term interaction refers to the modifier effect that two or more variables (e.g. variable 1 x variable 2) have (together or the interaction effect) in the outcome measured.

Line 282 "To avoid bias". This clustering may indeed induce bias. Change to "due to low detection rates for some viruses, we combined....".

What bias could the study have? There is no addressing of bias in the methods, results or discussion.

Line 290 change "virus interaction" for "virus correlation"

292: Mathematical or statistical models?

293: Indicate what type of regression was used I assume logistic regression model

294: Including known confounders into the model does not testing for confounding itself in the study.

Age
Gender
Season

Are included in the model to control "confounding". How was confounding assessed? Please clarify.

Was season modeled as a continuous variable? If there are four possible seasons why is there only one term "season" in the model?

297 "interactive items" are in fact what is known in epidemiology as interaction.

306 "Y" would indicate the odds of detection, not the prevalence rate.

There are no viral interaction terms included in the logistic regression hence it is not clear how "driver factors on virus interactions" (line 292) were measured. Perhaps better explanation in the methods, including reference in which the methods have been used to estimate virus interactions could help.

Results:

What was the number of samples tested per year? And was there any difference between years by age group. Include the "n" number of patients tested by year in any of the tables or plots presented to illustrate a balanced sample across the study period.

Figure 1: Line 484 and 487: Change “monthly prevalence” for “Monthly detection rate” Be consistent across the paper and use “detection rate” instead of “prevalence”

Line 131: Change “Interaction” for “Correlation” and be consistent through the paper to avoid confusion when measuring correlation based on the Spearman’s rank correlation test.

Figure 3: Line 504 and 509 as panel B of Figure 3 . Change “Virus interactions” for “Virus correlations”

Figure 4: Line 522 and panel A of Figure 4. Change interaction for correlations.

This figure needs WAY BETTER explanations of the results observed and the methods used to obtain the illustrated estimates. Panel A: Is the Y axis on the left representing the Spearman correlation coefficients illustrated in S2 overtime? Or from what model result is the heat mat plot. What is the percent illustrated on the right side of the heat map? What is the unit for the density scale on the right of the heat map. What are the six dotted lines over time in the heat map representing? Panel B: Are the colors representing something in the panel?

Include the number of patients included per year. How does the number of patients tested affect the correlation heat map observed?

Line 152-162: Confounding:

It is not clear through the methods or results how was confounding assessed. There is one logistic regression model (line 304) established to estimate the odds of virus (Y) detection given three exposure variables (age (three groups), gender (two groups), season (four groups) and one interaction or effect modifier term in the model (gender x age group).

The authors claim to have 6 variables that will yield 6 coefficients. In reality there are only 3 variables and one interaction term in the model. Age, gender, and season will not provide independent coefficients for each group because they depend on the other levels of each categorical variable.

Please clarify how these variables were included in the model, and what was the reference group used for each variable. Include the significance (p values and statistical test used) to assess the significance of each variable showed in table S4. Reduced vs full models would be useful.

Line 158: It is not clear how does Fig4B, Table S4 and S5 showed that seasonality contributed no effect on virus interactions.

Line161: What result (or model) supports the significance of 0.15 vs 0.10) and all other difference reported up to line 164.

Line 300: Explain the “correlation coefficients” obtained and illustrated in Table S4. It is not clear. Indicate the significance for each variable. And the test used to compare the full versus reduced model to assess confounding. Was season modeled as continuous variable (1, 2, 3, 4) instead of a categorical variable with four groups?

Response to Reviewers

MS title: Statistical analysis of common respiratory viruses reveals the binary of virus-virus interaction

MS No.: Spectrum00019-23-R1

Response to Reviewer #1

Understanding the interaction and interference between epidemics of respiratory viruses is conducive to the formulation of respiratory virus infection control strategies and risk assessment. There are few studies in this field. In this study, the author systematically characterized the interactions between respiratory viruses through mathematical modeling based on the data of multi-pathogen detection. The results showed there are stable interactions among respiratory viruses at population level, which are season-independent. The asynchronous interference between influenza virus and β human coronaviruses was explored, which significantly delayed the peak of β human coronaviruses epidemic during spring and summer.

This is a long-term and large-sample prospective study, with enrolling 14,426 patients suffered from acute respiratory infection (ARI) in Beijing, China during 2005-2015. Therefore, the results of this paper have good reference value, although as the author said, the article still has some limitations.

Minor revision

R1.1 In the discussion, the author should analyze the result: "The interaction between IFVs and HCoVs- β was significant in spring and summer, but not in autumn and winter". Why there are different results in different seasons?

RE: We appreciate your suggestion, based on which we have enriched the discusses of the results. In our study, the peaking time of HCoVs- β was mainly in summer, with 7-fold frequencies compared with that of winter. The overall detection rate of HCoVs- β in autumn and winter was only 0.6% and 0.1%, while 19.5% and 30.0% in the case of IFV-A at that time. By Spearman's rank correlation analysis, we obtained significant negative correlations between IFV-A and HCoVs- β in spring and

summer. We also noticed a similar tendency of the two viruses in autumn and winter. The low detection rate in the period may have influenced the interaction analysis. We have added this part to the discussion. Please refer to lines 227-230 in the revised manuscript.

R1.2 Another limitation is that children's cases are not included. This should be added to the limitation analysis of the article.

RE: We have added the lacking of children's cases as one of the limitations in the *Discussion* section. Please refer to lines 243-244 in the revised manuscript.

Response to Reviewer #2

R2.1 In the manuscript “Mathematical analysis of common respiratory viruses reveals the binary of virus-interactions and interference” Zhang et al., report a fascinating interaction of human respiratory viruses based on a prospective laboratory-based study with 14426 patients with acute respiratory infection (ARI) in Beijing between 2005 and 2015. Most conclusions are supported by the authors’ findings. However, several clarifications regarding the methods and interpretation must be addressed to make the paper clearer and stronger. The majority of the manuscript is well written, but a final English proofread is recommended to identify minor grammatical errors prior publication.

RE: Many thanks for these comments. We have asked a native English speaker to edit the manuscript as suggested.

R2.2 I believe the proposed models are more epidemiological or statistical than mathematical. Consider adjusting the title to “Statistical analysis of...”.

RE: Thanks for the suggestion! The title has been updated to reflect the nature of the statistical analysis, and the new title is “Statistical analysis of common respiratory viruses reveals the binary of virus-virus interaction.”

R2.3 It would be useful to better define virus “interaction”, “interference”, and “correlation” in the methods sections, and use these terms consistently through the paper to avoid confusion. Even in the title the authors use “interaction and interference” without giving clarity of its definition for the purpose of the study.

RE: We thank reviewer’s helpful suggestion. We have added the definitions of virus “interaction”, “interference” and “correlation” in the section of *Materials and Methods*. These definitions are inherited from previously published literatures. “Virus interaction” refers to when multiple pathogens cocirculate in same time and space which can lead to competitive or cooperative forms between virus–virus (*Nickbakhsh S et al. Proc Natl Acad Sci U S A, 2019; Bosch A et al. PLoS Pathog, 2013; Shaman J et al. Science, 2020*). “Virus interference” describes the situation whereby infection with one virus limits infection and replication of a second virus (*Laurie K et al. J Infect Dis, 2018; Chan K et al. J Infect Dis, 2018; Schultz-Cherry S et al. J Infect Dis, 2015; Laurie K et al. J Infect Dis, 2015*). “Correlation” indicates statistical evidence based on time series data of viral epidemiology. The correlation results include the influence of other factors and the genuine interactions between viruses. Both virus interaction and virus interference could result in correlations in data patterns.

We have incorporated the above explanations and added the citations in the revised manuscript. Please refer to the title and lines 290-296 in the revised manuscript.

R2.4 “Interaction” in statistics or epidemiology differ of what the authors call virus “interactions” or correlations. It seems that these terms (interactions, interference and correlations) are used across the manuscript synonymously but is not clear for the reader what comes from the interaction terms in model 1 (line 304), the Spearman’s rank test (line 281), and what yield the density heat map of interactions shown in Figure.

RE: We have added the definitions of these terms in *Materials and Methods* section. Please refer to lines 290-296 in the revised manuscript as R2.3.

We have modified model 1 to multivariable vector autoregressive model to infer genuine viral interactions, differ from simple correlations. Please refer to lines 309-326

and 170-181 for more details of the model.

Spearman's rank test is used to study the correlation among viruses using month detection rate. We have revised the interaction to correlation analysis. Please refer to line 289 and 294.

The input data for the density heat map is the correlation between each of the two viruses per year to explore the temporal distribution of the correlation. Relevant descriptions have been added to the revised manuscript. We also modified “interactions” to “correlations”. Please refer to lines 165-169 and 537-544 in revised manuscript.

R2.5 A better explanation of the models used to estimate interactions and correlation may help the reader to follow smoothly the study results.

RE: We thank the reviewer for the helpful suggestions. We added more details of the models in the section of *Materials and Methods*. Please refer to lines 289-307, 308-326 and lines 327-343 in the revised manuscript.

Specifically, Spearman simple correlation analysis was used to analyze the correlations among viruses at the population level, evaluated as the monthly detection rates. Vector autoregressive model was used to identify genuine viral interactions from simple correlations based on time series data from multiple contemporaneous viruses adjusted for confounding factors such as age, gender and season. Susceptible-Infectious-Recovered (SIR) model was used to analyze the effects of virus interaction on the epidemic of viruses.

Methods sections:

R2.6 Line 249: clarify if the remaining patients (after the 5-10 recruited) where not included in the study.

RE: In case of sampling bias, we only included the first 5-10 patients met the criteria on each day in this study. The remaining patients were not included. We have added this declaration. Please refer to lines 258-260 in the revised manuscript.

R2.7 Line 268: Detection rate. Stay consistently through the paper and do not use “prevalence” is not a prevalence study. Legend for figure 1 (line 485) should not read “Monthly prevalence”.

RE: We have revised the manuscript by replacing “prevalence” with “detection rate”. Please refer to line 279 and legend for Figure 1 (lines 503-506) in the revised manuscript.

R2.8 Line 278: Is a correlation analysis hence a correlation coefficient is obtained. Not an interaction analysis. Change the subtitle of this section. Given the logistic regression model used, the term interaction refers to the modifier effect that two or more variables (e.g variable 1 x variable 2) have (together or the interaction effect) in the outcome measured.

RE: We have revised the subtitle to “Virus correlation analysis”. Please refer to line 289 in the revised manuscript. In logistic regression model, the term interaction indeed refers to the modifier effect that two or more variables have in outcome. In addition, we have modified model 1 to multivariable vector autoregressive model to infer genuine viral interactions. Please refer to lines 309-326 and 170-181 for more details of the model.

R2.9 Line 282 “To avoid bias”. This clustering may indeed induce bias. Change to “due to low detection rates for some viruses, we combined....”. What bias could the study have? There is no addressing of bias in the methods, results or discussion.

RE: We have changed it to “Due to low detection rates for some viruses, we combined some viruses according to the biological properties”. Please refer to lines 298-299 in the revised manuscript. We have added the possible bias from virus subtypes clustering in the *Discussion* section (lines 236-240).

R2.10 Line 290 change “virus interaction” for “virus correlation”.

RE: We have revised it as suggested. Please refer to line 306 in the revised manuscript.

R2.11 292: Mathematical or statistical models?

RE: Thanks for pointing this out. It is actually a statistical model, instead of a “mathematical model”. We have changed the paper title and the descriptions in the manuscript. Please refer to line 308 in the revised manuscript.

R2.12 293: Indicate what type of regression was used I assume logistic regression model.

RE: We have modified model 1 to multivariable vector autoregressive model to infer genuine viral interactions. Please refer to lines 309-326 in the revised manuscript for more details and R2.5.

R2.13 294: Including known confounders into the model does not testing for confounding itself in the study. Age, Gender, Season...Are included in the model to control “confounding”. How was confounding assessed? Please clarify.

RE: We have modified model 1 to multivariable vector autoregressive model to infer genuine viral interactions. In this model setup, the confounding factors are assumed to be additive. The multivariate vector autoregressive approach allows us to control the confounding factors as exogenous variables, and the genuine correlations among the virus are therefore captured by the covariance matrix among the residuals in the model. The model focuses on effects from the lag of the pathogen itself and the impulse response and lag of other viral variables. Ten viruses (including IFV-A, IFV-B, HPIV 2/4, HPIV 1/3, HCOVs- β , Adv, HCOVs- α , hMPV, RSV and picoRNA) were considered as endogenous variables in this model. Age, gender and season have been proved to be the main factors affecting the virus detection in other studies (Zhang L et al. BMC Infect Dis, 2023; Ren L et al. Clin Microbiol Infect, 2009; Yu J et al. BMC Infect Dis, 2018). Therefore, age, gender and season were considered as exogenous variables to control the confounding effects. Please refer to lines 309-326 in the revised manuscript for more details.

R2.14 Was season modeled as a continuous variable? If there are four possible seasons why is there only one term “season” in the model?

RE: We thank reviewer for the kind suggestions. We have modified model 1 to multivariable vector autoregressive model to infer genuine viral interactions and season has been modeled as a categorical variable with values of 1, 2, 3, 4 in this model.

R2.15 297 “interactive items” are in fact what is known in epidemiology as interaction.

RE: We have modified model 1 to multivariable vector autoregressive model and the term “interactive items” is no longer needed. Please refer to lines 309-326 in the revised manuscript.

R2.16 306 “Y” would indicate the odds of detection, not the prevalence rate.

RE: Revision has been made as suggested. In the revised model, the Y values are calculated as the ln of the odds ratio. Given that in some months, the detection rate is 0 for a virus, the odds ratio was added value 1 before taking the ln-transformation. Please refer to lines 314-317 in the revised manuscript for details.

R2.17 There are no viral interaction terms included in the logistic regression hence it is not clear how “driver factors on virus interactions” (line 292) were measured. Perhaps better explanation in the methods, including reference in which the methods have been used to estimate virus interactions could help.

RE: We have modified model 1 to multivariable vector autoregressive model to infer viral interactions. The multivariate VAR model allows us to control two other possible factors that may lead to correlations, (1) inter-temporal auto-correlations for the same virus over time; (2) the co-occurrence of more than one virus among the same demographics, such as age and gender. After controlling for the above possible factors, the additional correlations among the virus is captured by the covariance matrix of the error term included in the multivariate VAR model. Detailed descriptions and references have been added in the manuscripts. Please refer to lines 309-326 and

170-181 for more details of the model.

Results:

R2.18 What was the number of samples tested per year? And was there any difference between years by age group. Include the “n” number of patients tested by year in any of the tables or plots presented to illustrate a balanced sample across the study period.

RE: The number of samples tested per year was added in Figure 4A. The distribution of samples by age group between years were similar. Please refer to lines 166-167 and Figure S3.

R2.19 Figure 1: Line 484 and 487: Change “monthly prevalence” for “Monthly detection rate” Be consistent across the paper and use “detection rate” instead of “prevalence”.

RE: Revision has been made as suggested. Please refer to lines 503-504 and 506 in the revised manuscript.

R2.20 Line 131: Change “Interaction” for “Correlation” and be consistent through the paper to avoid confusion when measuring correlation based on the Spearman’s rank correlation test.

RE: Revision has been made as suggested. Please refer to line 143 in the revised manuscript.

R2.21 Figure 3: Line 504 and 509 as panel B of Figure 3. Change “Virus interactions” for “Virus correlations”.

RE: We have revised it as suggested. Please refer to lines 519 and 524 in the revised manuscript.

R2.22 Figure 4: Line 522 and panel A of Figure 4. Change interaction for correlations. This figure needs WAY BETTER explanations of the results observed and the methods used to obtain the illustrated estimates.

Panel A: Is the Y axis on the left representing the Spearman correlation coefficients illustrated in S2 overtime? Or from what model result is the heat map plot. What is the percent illustrated on the right side of the heat map? What is the unit for the density scale on the right of the heat map. What are the six dotted lines over time in the heat map representing?

Panel B: Are the colors representing something in the panel? Include the number of patients included per year. How does the number of patients tested affect the correlation heat map observed?

RE: We have changed “interactions” to “correlations” in panel A of Figure 4. Please refer to line 538 in the revised manuscript. The panel B and C have been modified as the model have been changed and detailed explanations have been added in the legends of Figure 4. Please refer to lines 537-549.

In Panel A, the Y axis on the left represented the Spearman correlation coefficients by year. The percentages shown on the right side of the heat map represent the mean, median, quartile and range of the correlation coefficients, and the changes of these values by year was represented as six dotted lines. The density bar on the right represents the density of the correlation coefficient distribution. There is no unit or the unit is the number of specific correlation coefficient value.

We also have added the annual number of tested cases in Figure 4A. Colors don't mean anything in the original panel B, they're random. Although the total number differs year by year, the composition of age groups is more balanced, so the impact on virus correlation may be limited. Please refer to lines 166-167 and Figure S3.

R2.23 Line 152-162: Confounding: It is not clear through the methods or results how was confounding assessed. There is one logistic regression model (line 304) established to estimate the odds of virus (Y) detection given three exposure variables

(age (three groups), gender (two groups), season (four groups) and one interaction or effect modifier term in the model (gender x age group)).

RE: We have modified model 1 to multivariable vector autoregressive model to infer genuine viral interactions. The model focuses on the effects from the lag of the pathogen itself and the impulse response and lag of other viral variables. Ten viruses (including IFV-A, IFV-B, HPIV 2/4, HPIV 1/3, HCOVs- β , Adv, HCOVs- α , hMPV, RSV and picoRNA) were considered as endogenous variables in this model. Age, gender and season have been proved to be the main factors affecting the virus detection rate in previous studies (*Zhang L et al. BMC Infect Dis, 2023; Ren L et al. Clin Microbiol Infect, 2009; Yu J et al. BMC Infect Dis, 2018*). Therefore, age, gender and season were considered as exogenous variables to control the effect from confounding factors. Please refer to lines 309-326 in the revised manuscript and also to R2.13 and R2.17.

R2.24 The authors claim to have 6 variables that will yield 6 coefficients. In reality there are only 3 variables and one interaction term in the model. Age, gender, and season will not provide independent coefficients for each group because they depend on the other levels of each categorical variable.

RE: We have modified model 1 to multivariable vector autoregressive model to infer genuine viral interactions. Age, gender and season were considered as exogenous variables in this model. The three confounding variables (age [four groups], gender [two groups], season [four groups]) are all categorical variable. The last group was used as the reference in the model. More details of the model used were added in the section of *Materials and Methods*. Please refer to lines 309-326 in the revised manuscript.

R2.25 Please clarify how these variables were included in the model, and what was the reference group used for each variable. Include the significance (p values and statistical test used) to assess the significance of each variable showed in table S4. Reduced vs full models would be useful.

RE: We have modified model 1 to multivariable vector autoregressive model to infer genuine viral interactions. In this model, ten viruses (including IFV-A, IFV-B, HPIV 2/4, HPIV 1/3, HCOVs- β , Adv, HCOVs- α , hMPV, RSV and picoRNA) were considered as endogenous variables, and age, gender and season were considered as exogenous variables. The last group was used as the reference in the model. Full model has been added in lines 309-326 in the revised manuscript.

R2.26 Line 158: It is not clear how does Fig 4B, Table S4 and S5 showed that seasonality contributed no effect on virus interactions.

RE: The model has been changed and Fig 4B, Table S4 and S5 were also modified. Vector autoregressive model has been used to control confounding factors including age, gender and season. Please refer to the revised manuscript.

R2.27 Line161: What result (or model) supports the significance of 0.15 vs 0.10) and all other difference reported up to line 164.

RE: As mentioned above, the model has been replaced. Please refer to lines 309-326 and 170-181 in the revised manuscript.

R2.28 Line 300: Explain the “correlation coefficients” obtained and illustrated in Table S4. It is not clear. Indicate the significance for each variable. And the test used to compare the full versus reduced model to assess confounding. Was season modeled as continuous variable (1, 2, 3, 4) instead of a categorical variable with four groups?

RE: We have modified the model to vector autoregressive model to infer genuine viral interactions adjusted confounding factors such as age, gender and season. Season was a categorical variable with four groups in our study. Table S4 was also changed. Full model has been added in *Materials and Methods* section. Please refer to lines 309-326 in the revised manuscript and also to R2.13, R.17, and R2.23.

June 9, 2023

Dr. Jianwei Wang
Institute of Pathogen Biology, Peking Union Medical College & Chinese Academy of Medical Sciences
9 Dong Dan San Tiao, Dongcheng District,
Beijing, Beijing 100730
China

Re: Spectrum00019-23R1 (**Statistical analysis of common respiratory viruses reveals the binary of virus-virus interaction**)

Dear Dr. Jianwei Wang:

Your manuscript has been accepted, and I am forwarding it to the ASM Journals Department for publication. You will be notified when your proofs are ready to be viewed.

Sincerely,

Sen Pei
Editor, Microbiology Spectrum

Journals Department
**Statistical analysis of common respiratory viruses reveals the binary**
**of virus-virus interaction**

Lulu Zhang¹ Ph.D†, Yan Xiao^{1,2} MS†, Zichun Xiang¹ Ph.D, Lan Chen¹ BS, Ying
Wang¹ MS, Xinming Wang¹ MS, Xiaojing Dong³ Ph.D*, Lili Ren^{1,2} Ph.D*, Jianwei
Wang^{1,2} Ph.D*

¹ Institute of Pathogen Biology, Chinese Academy of Medical Sciences & Peking
Union Medical College, Beijing 100730, P.R. China

² Key Laboratory of Respiratory Disease Pathogenomics, Chinese Academy of
Medical Sciences and Peking Union Medical College, Beijing 100730, P.R. China

³ Santa Clara University, 500 El Camino Real Santa Clara, CA 95053

† These authors contributed equally to this work as first authors.

* These authors contributed equally to this work as senior authors.

**Corresponding to:**

Dr. Jianwei Wang

No.9 Dong Dan San Tiao, Dongcheng District, Beijing 100730, P. R. China

Tel/Fax: 86-10-67828516

E-mail: wangjw28@163.com

Dr. Lili Ren

No.9 Dong Dan San Tiao, Dongcheng District, Beijing 100730, P. R. China

Tel/Fax: 86-10-67828516

E-mail: renliliipb@163.com

Prof. Xiaojing Dong

Santa Clara University, 500 El Camino Real Santa Clara, CA 95053

Tel/Fax: 408-554-5721

E-mail: xdong1@scu.edu

[revised manuscript text omitted]